# Myosin-based regulation of twitch and tetanic contractions in mammalian skeletal muscle

**Cameron Hill\*, Elisabetta Brunello, Luca Fusi, Jesús G Ovejero, Malcolm Irving**

Randall Centre for Cell & Molecular Biophysics, New Hunt's House, Guy's Campus, King's College London, London, United Kingdom

**Abstract** Time-resolved X-ray diffraction of isolated fast-twitch muscles of mice was used to show how structural changes in the myosin-containing thick filaments contribute to the regulation of muscle contraction, extending the previous focus on regulation by the actin-containing thin filaments. This study shows that muscle activation involves the following sequence of structural changes: thin filament activation, disruption of the helical array of myosin motors characteristic of resting muscle, release of myosin motor domains from the folded conformation on the filament backbone, and actin attachment. Physiological force generation in the 'twitch' response of skeletal muscle to single action potential stimulation is limited by incomplete activation of the thick filament and the rapid inactivation of both filaments. Muscle relaxation after repetitive stimulation is accompanied by a complete recovery of the folded motor conformation on the filament backbone but by incomplete reformation of the helical array, revealing a structural basis for post-tetanic potentiation in isolated muscles.

## Introduction

The unitary contractile response of skeletal muscle to an action potential in the muscle cell membrane—the twitch—is triggered by a transient increase in intracellular calcium concentration ($[Ca^{2+}]_i$), leading to calcium binding to troponin in the actin-containing thin filaments. This, in turn, initiates a change in the thin filament structure in which tropomyosin moves from its blocking position in resting muscle, allowing myosin head or motor domains from the thick filaments to bind to actin and generate force (*Gordon et al., 2000*). The duration of the $[Ca^{2+}]_i$ transient is much briefer than the mechanical response in the twitch, but peak $[Ca^{2+}]_i$ is an order of magnitude larger than the dissociation constant of the $Ca^{2+}$ regulatory sites on troponin, which become fully occupied with a delay of less than 1 ms after peak $[Ca^{2+}]_i$ in mammalian muscle at 28°C (*Baylor and Hollingworth, 2003*). The movement of tropomyosin in the thin filament is also much faster than force development, with a half-time of about 5 ms as determined by time-resolved X-ray diffraction in amphibian fast-twitch muscles at 22°C, conditions in which the twitch duration is about 100 ms (*Kress et al., 1986*).

Although the thin filaments are rapidly and fully activated in the twitch, the peak force in the twitch is much less than that produced at full activation, as produced in intact muscles by repetitive high-frequency stimulation in a tetanus. The force at the tetanus plateau is about four times larger than that at the peak of the twitch in mammalian fast-twitch muscles at near-physiological temperature. Tetanic stimulation increases the duration, but not the peak amplitude, of the $[Ca^{2+}]_i$ transient or the peak occupancy of the $Ca^{2+}$ regulatory sites on troponin compared to that in the twitch (*Baylor and Hollingworth, 2003*). It follows that the $[Ca^{2+}]_i$ transient provides a 'start' signal for contraction but does not control either the amplitude or the time course of the twitch in skeletal muscle.

**\*For correspondence:**
cameron.hill@kcl.ac.uk

**Competing interests:** The authors declare that no competing interests exist.

What molecular mechanisms do then determine the strength and speed of the twitch? The rate of force development is effectively limited by the rate of binding of myosin motors to thin filaments, as determined from the time course of instantaneous stiffness during force development in tetani of amphibian muscle at 4°C after taking filament compliances into account (*Brunello et al., 2006*; *Fusi et al., 2014a*). The question then becomes 'What determines the rate at which myosin motors bind to thin filaments?' One possibility, suggested by mechanical and biochemical studies on demembranated muscle fibres (*Brenner and Eisenberg, 1987*; *Goldman, 1987*), is that a rate-limiting biochemical step in the actin-bound motor controls the transition between weakly and strongly actin-bound states, the latter being detected by the instantaneous stiffness measurements and required for active force generation.

An alternative or additional possibility emerged more recently from accumulating evidence that the thick filament also has a regulatory role. In resting muscle, the myosin motors are not available to interact with actin and generate contraction because they are folded back against the filament backbone in a helical array stabilised by interactions with other myosin motors and other thick filament components, including myosin-binding protein-C (MyBP-C) and titin (*Irving, 2017*; *Woodhead et al., 2005*). This thick filament 'off' state, sometimes described as the 'super-relaxed' state, minimises ATP hydrolysis and the associated whole-body metabolic cost in resting muscle (*Stewart et al., 2010*), but its existence raises the question of how the *thick* filament, which does not appear to have a direct calcium signalling mechanism, is activated by electrical stimulation. Part of the answer to that question was provided by the discovery that the thick filament can be directly activated by mechanical stress (*Fusi et al., 2016*; *Linari et al., 2015*). More generally, X-ray studies of the rising phase of the tetanus in amphibian muscle at 4°C showed that switching on the thick filaments is faster than force development (*Reconditi et al., 2011*). However, the extent to which the thick filament is switched on during a twitch, or controls the strength and dynamics of the twitch, has not been investigated.

Finally, the mechanisms that control the strength and speed of contraction in the twitch cannot be separated from those that control relaxation. From the perspective of $[Ca^{2+}]_i$, and thin filament regulation, almost the entire time course of the twitch force occurs after $[Ca^{2+}]_i$ has returned almost to baseline, although there may be a small, slow tail in the $[Ca^{2+}]_i$ transient (*Caputo et al., 1994*; *Konishi, 1998*), which, even at a very small fraction of the peak, could significantly slow $Ca^{2+}$ dissociation from troponin. Moreover, the thin filament may not switch off immediately after $Ca^{2+}$ has dissociated from troponin because myosin motors that remain bound to actin would hold tropomyosin in its 'on' position in the thin filament. Changes in the length of intact muscle fibres during relaxation alter the $[Ca^{2+}]_i$ transient (*Cannell, 1986*; *Caputo et al., 1994*), implying that the extent of $[Ca^{2+}]$ binding to the thin filament during relaxation is sensitive to muscle length or load. Thick filament mechano-sensing (*Linari et al., 2015*) could also contribute to maintaining the 'on' state of the thick filament during relaxation. In isolated myofibrils and in single fibres from amphibian muscle, relaxation can be clearly separated into a slow, almost linear phase in which sarcomere lengths remain almost constant ('isometric relaxation') followed by mechanical yielding and rapid 'chaotic relaxation' in which the remaining force is rapidly lost (*Brunello et al., 2009*; *Poggesi et al., 2005*). The extent to which this phenomenon contributes to the time course of relaxation in intact, fast-twitch mammalian muscle fibres at more physiologically relevant temperatures is unknown.

Here we used time-resolved X-ray diffraction of intact, electrically stimulated, extensor digitorum longus (EDL) muscles of mice to address these fundamental gaps in understanding the mechanisms that determine the amplitude and complete time course of the twitch in mammalian muscle, focusing on the X-ray reflections that report the structure and regulatory state of the thick filaments and the conformations of the myosin motors. The results are complementary to those of a pioneering time-resolved X-ray study of twitch and tetanus in amphibian muscle that focused on the thin filament-based X-ray reflections (*Kress et al., 1986*). All of the present measurements on mouse EDL muscle were made at 28°C, a temperature at which the muscles contract reproducibly as required for signal averaging in X-ray experiments. Moreover, the time course of the $[Ca^{2+}]_i$ transient and its binding to troponin has been characterised in mouse EDL at 28°C (*Baylor and Hollingworth, 2003*), and the peak force is very close to that at the mammalian body temperature (*Caremani et al., 2019*). Most importantly, the off structure of the thick filament is preserved in mammalian muscle at this temperature (*Caremani et al., 2021*), in contrast with nearly all the published studies of thin filament regulation, for example, those using pCa titrations with steady-state Ca buffering in skinned

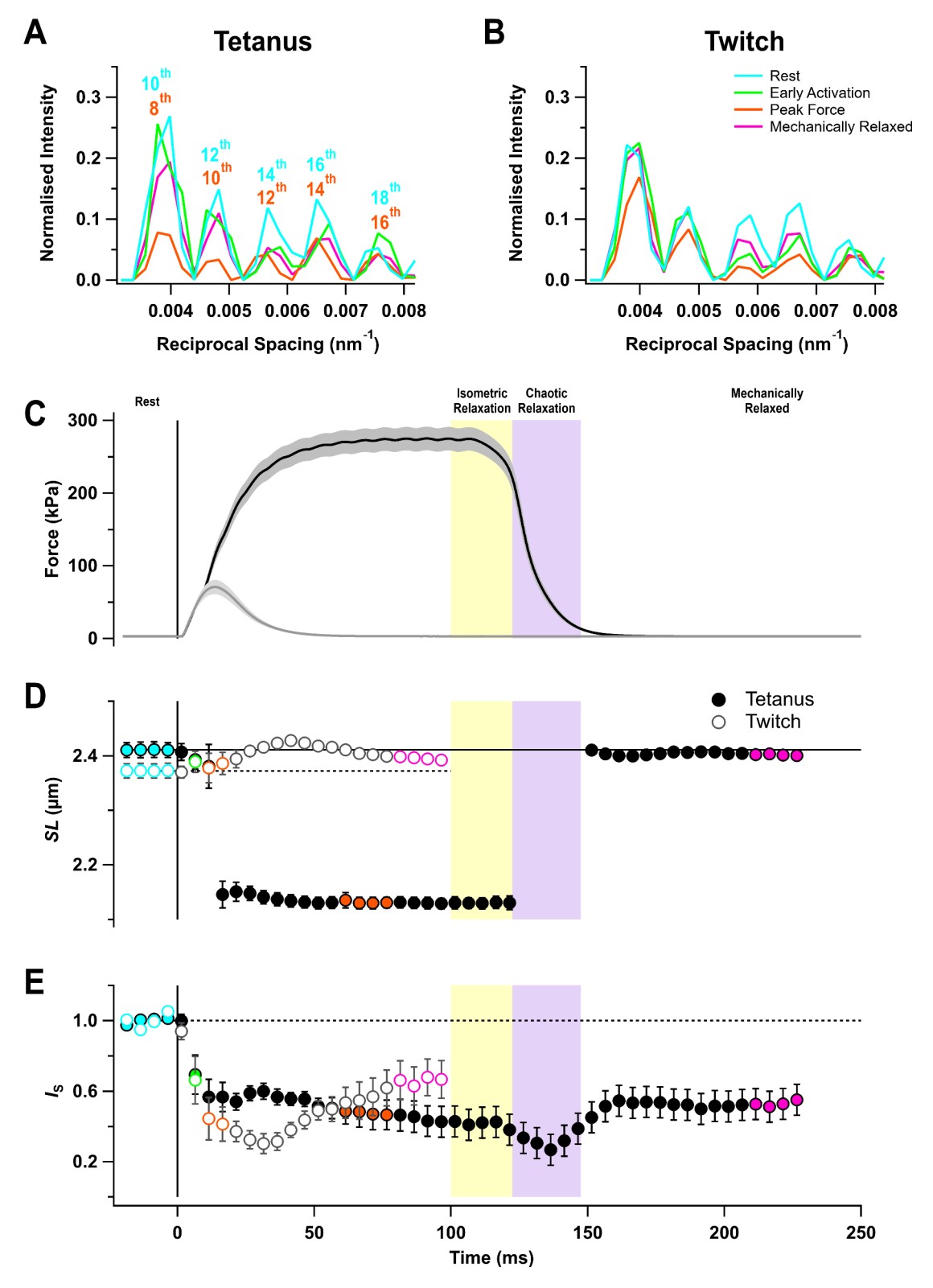

**Figure 1.** Force and sarcomere length changes in fixed-end twitch and tetanus. (**A**) Ultra-low-angle X-ray reflections from the sarcomere periodicity with indicated orders of the fundamental sarcomere length repeat in a tetanus. Cyan, rest; green, early activation; orange, tetanus plateau; magenta, mechanically relaxed. (**B**) Corresponding results for the twitch, with orange denoting the peak force in the twitch. (**C**) Time course of changes in force with SEM indicated by grey shading. (**D**) Sarcomere length (*SL*). (**E**) Intensity of sarcomere reflections ($I_S$) determined from the average intensities of the

*Figure 1 continued on next page*

*Figure 1 continued*

third and fourth peaks in the region shown in (**A**) and (**B**) normalised by the mean resting value. Filled and open symbols in (**D**) and (**E**) denote tetanus and twitch, respectively; coloured symbols denote the time periods used to calculate the profiles in (**A**) and (**B**); error bars denote SEM for n = 5 muscles for tetanus and n = 4 muscles for twitch. Yellow- and purple-shaded panels denote isometric and chaotic relaxation. Black horizontal continuous and dashed lines denote resting values.

The online version of this article includes the following source data for figure 1:

**Source data 1.** An excel file containing the data for individual muscles from which the mean and SEM shown in *Figure 1* were calculated.

muscle fibres, which were made at a relatively low temperature in order to minimise the irreversible effects of sustained high levels of activation at a high temperature. The role of thick filament regulation was inadvertently excluded from such experiments, because the thick filaments were already switched on at the low temperature used, even in the absence of calcium.

## Results

### Sarcomere length changes during twitch and tetanic contractions at a fixed muscle length

The I22 beamline at the Diamond Light Source allows the X-ray diffraction pattern from intact muscles to be collected over a wide range of diffraction angles, corresponding to structural periodicities from about 300 to 5 nm. The ultra-low-angle X-ray reflections from the sarcomere periodicity could therefore be recorded at the same time and from the same region of the muscle as the conventional small-angle reflections that give information about the structure of the thick filaments and the conformation of the myosin motors. Five sarcomere reflections were observed, corresponding to even orders of the sarcomere repeat from the $10^{th}$ to the $18^{th}$ order in resting muscle (*Figure 1A,B*, cyan) (*Bordas et al., 1987*; *Reconditi et al., 2014*). Sarcomere length in the resting muscles was about 2.4 μm (*Figure 1D*; *Table 1*).

The sarcomere reflections became much weaker during stimulation, but sarcomere length in the tetanus reached a roughly constant value of 2.13 μm from about 20 ms after the first stimulus (*Figure 1D*, filled circles; *Table 1*), indicating that the sarcomeres in the central region of the muscle

**Table 1.** Force, sarcomere length, and X-ray parameters at rest and peak force in twitch and tetanus.

Rest, average of four frames from −18.5 ms to −3.5 ms for twitch and tetanus; PF tetanus, average of four frames from 61.5 ms to 76.5 ms; PF twitch, average of two frames at 11.5 ms and 16.5 ms. All X-ray intensities have been normalised by their resting values. Mean ± SEM for n = 5 for tetanus and n = 4 for twitch. *SL*, sarcomere length.

| | Tetanus | | Twitch | |
|---|---|---|---|---|
| | **Rest** | **PF** | **Rest** | **PF** |
| Force (kPa) | - | 273 ± 16 | - | 68 ± 10 |
| *SL* (μm) | 2.41 ± 0.01 | 2.13 ± 0.01 | 2.37 ± 0.01 | 2.38 ± 0.03 |
| $d_{1,0}$ (nm) | 35.32 ± 0.11 | 35.74 ± 0.17 | 35.07 ± 0.15 | 35.44 ± 0.25 |
| $I_{1,1}/I_{1,0}$ | 0.41 ± 0.02 | 1.77 ± 0.14 | 0.39 ± 0.01 | 0.93 ± 0.05 |
| $I_{AL1}$ | 1 | 2.66 ± 0.43 | 1 | 0.74 ± 0.11 |
| $I_{ML1}$ | 1 | 0.09 ± 0.03 | 1 | 0.28 ± 0.04 |
| $A_{ML1}$ | 1 | 0.36 ± 0.06 | 1 | 0.53 ± 0.04 |
| $A_{AL1}$ | 1 | 1.61 ± 0.13 | 1 | 0.84 ± 0.07 |
| $I_{M3}$ | 1 | 3.21 ± 0.26 | 1 | 0.69 ± 0.13 |
| $S_{M3}$ (nm) | 14.339 ± 0.003 | 14.535 ± 0.002 | 14.344 ± 0.004 | 14.425 ± 0.004 |
| $I_{M6}$ | 1 | 1.12 ± 0.26 | 1 | 0.81 ± 0.15 |
| $S_{M6}$ (nm) | 7.173 ± 0.002 | 7.283 ± 0.001 | 7.173 ± 0.002 | 7.241 ± 0.002 |

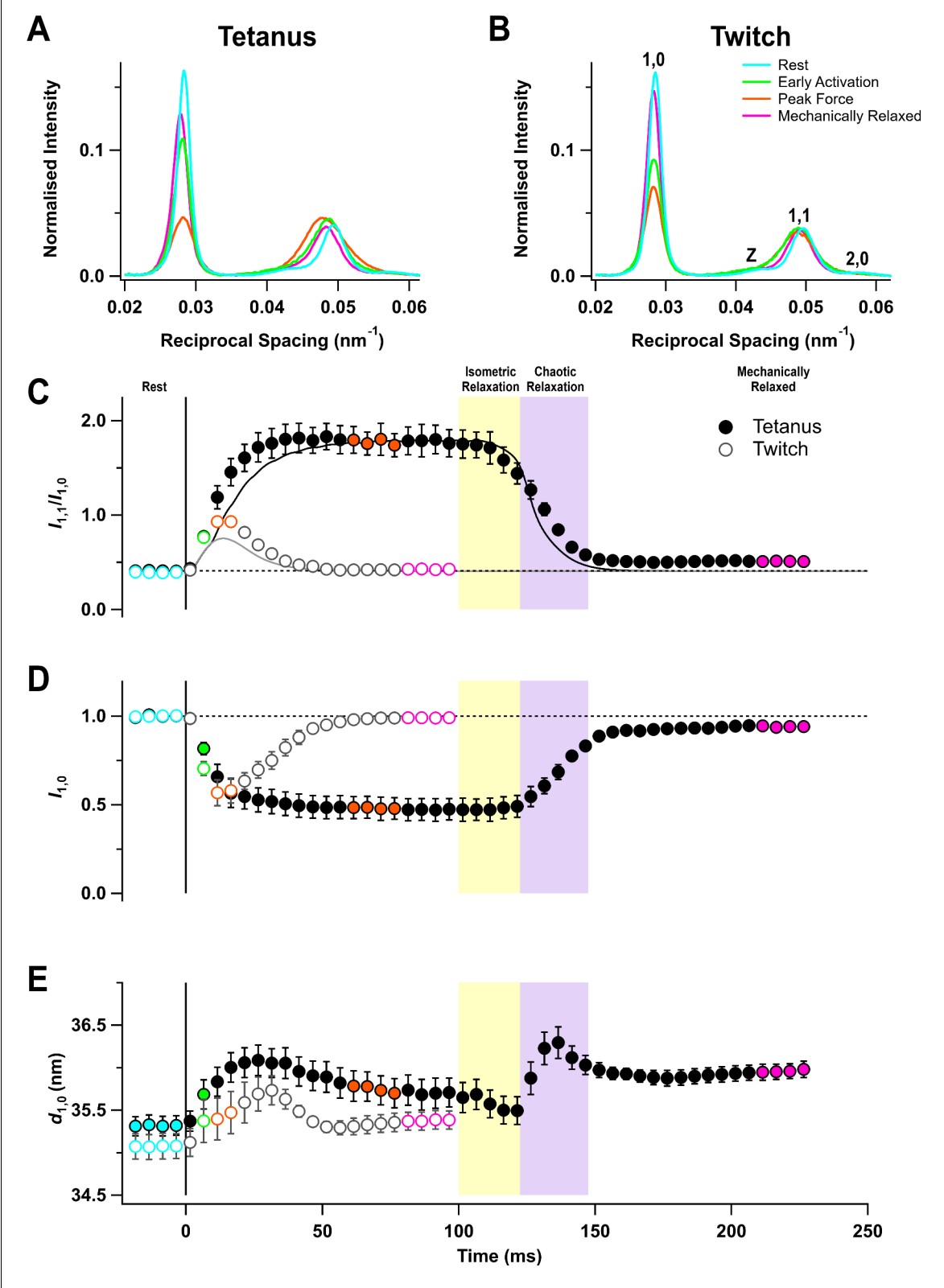

**Figure 2.** Equatorial X-ray reflections. (**A**) Distribution of intensity along the equator of the diffraction pattern, perpendicular to the muscle long axis, in different phases of a tetanus. Cyan, rest; green, early activation; orange, tetanus plateau; magenta, mechanically relaxed. (**B**) Corresponding results for the twitch, with orange denoting the peak force in the twitch. (**C**) Time course of the ratio of the intensities of (1,0) and (1,1) reflections ($I_{1,1}/I_{1,0}$) superimposed on the force in the tetanus (black) and twitch (grey). (**D**) Time course of the intensity of (1,0) reflection ($I_{1,0}$) normalised by the mean

*Figure 2 continued on next page*

*Figure 2 continued*

resting value. (E) Time course of the filament lattice spacing parameter $d_{1,0}$. Filled and open symbols in (C), (D), and (E) denote tetanus and twitch respectively; coloured symbols denote the time periods used to calculate the profiles in (A) and (B); error bars denote SEM for n = 5 muscles for tetanus and n = 4 muscles for twitch. Yellow- and purple-shaded panels denote isometric and chaotic relaxation. Black horizontal dashed lines denote resting values.

The online version of this article includes the following source data for figure 2:

**Source data 1.** An excel file containing the data for individual muscles from which the mean and SEM shown in *Figure 2* were calculated.

from which the X-ray signals were recorded had shortened by about 12%, presumably by stretching the tendons or compliant regions at the ends of the muscle fibres. This sarcomere length was maintained for about 20 ms after the last stimulus while force decreased to about 80% of its plateau value (*Figure 1C,D*, filled circles). Sarcomere lengths then suddenly became inhomogeneous, and it was not possible to assign sarcomere reflections or determine sarcomere lengths until ~50 ms after the last stimulus when force relaxation was more than 95% complete. Relaxation of mouse EDL muscle from 100-ms tetani, therefore, exhibits clear 'isometric' and 'chaotic' phases similar to those described previously in isolated myofibrils and single fibres from amphibian muscle (*Brunello et al., 2009*; *Poggesi et al., 2005*). By 50 ms after the last stimulus, the sarcomere length had recovered to close to its pre-stimulus value, although sarcomere heterogeneity had not recovered as indicated by the continued low intensity of the sarcomere reflections (*Figure 1E*). There appeared to be little sarcomere shortening in the twitch (*Figure 1D*, open circles), although the very low intensity of the sarcomere reflections (*Figure 1E*, open circles) suggests that sarcomere lengths became inhomogeneous, as during the first 20 ms of the tetanus.

## Equatorial X-ray reflections

The equator of the X-ray diffraction pattern from resting muscles, reporting the diffraction profile perpendicular to the long axis of the muscle and filaments (*Figure 2A,B*, cyan), shows the (1,0 , 1,1) and (2,0) reflections from the hexagonal lattice of thick and thin filaments and the Z reflection from the square lattice of thin filaments at the Z-disc. The (1,0) reflection became much weaker on activation in both the tetanus (*Figure 2A*, orange) and the twitch (*Figure 2B*), signalling motion of the myosin motors away from the thick filament backbone (*Haselgrove and Huxley, 1973*). The (1,1) reflection became broader on activation and its integrated intensity increased (*Figure 2A,B*, orange),

**Table 2.** Half-times of force and X-ray parameters during activation and relaxation.

Half-times ($t_{1/2}$) for the rising phase of the tetanus (activation) and for relaxation following tetanus and twitch. $t_{1/2}$ values for X-ray parameters were determined by fitting sigmoidal curves to the 5-ms time interval data. $t_{1/2}$ values for tetanus activation and twitch relaxation are reported with respect to the first stimulus at time 0 and those for tetanus relaxation with respect to the last stimulus at 100 ms. Rows are sorted by $t_{1/2}$ for tetanus activation, with the fastest signals at the top. No clear change in $A_{AL1}$ was observed during the twitch. $L_{M3}$, $M_{M3}$, and $H_{M3}$ are, respectively, the fractional intensities of the low-, mid- and high-angle peaks of the M3 reflection in *Figure 5E*. n = 5 muscles for tetanus and n = 4 muscles for twitch. Data presented are mean ± SEM. *A significant difference in $t_{1/2}$ for the X-ray parameter with respect to force in that phase. *p<0.05; **p<0.001; ***p<0.0001.

|  | Tetanus activation $t_{1/2}$ (ms) | Tetanus relaxation $t_{1/2}$ (ms) | Twitch relaxation $t_{1/2}$ (ms) |
|---|---|---|---|
| $S_{M6}$ (nm) | 8.1 ± 0.7*** | 32.4 ± 0.9* | 25.7 ± 0.6 |
| $A_{ML1}$ | 8.6 ± 0.9*** | 41.8 ± 2.6** | 29.4 ± 2.3 |
| $M_{M3}$ | 8.8 ± 0.5*** | 39.5 ± 1.4*** | 25.3 ± 1.7 |
| $S_{M3}$ (nm) | 10.9 ± 1.0** | 33.6 ± 1.2* | 20.1 ± 0.8* |
| $L_{M3}$ | 11.0 ± 0.6** | 37.7 ± 1.1*** | 25.8 ± 2.8 |
| $I_{1,1}/I_{1,0}$ | 11.3 ± 0.7** | 30.3 ± 0.8* | 28.4 ± 0.8* |
| $A_{AL1}$ | 15.3 ± 1.2 | 23.4 ± 1.9 | - |
| Force (kPa) | 17.0 ± 0.7 | 27.3 ± 0.6 | 24.4 ± 1.3 |
| $A_{M3}$ | 21.4 ± 1.3* | 20.9 ± 0.7** | 40.0 ± 1.7** |

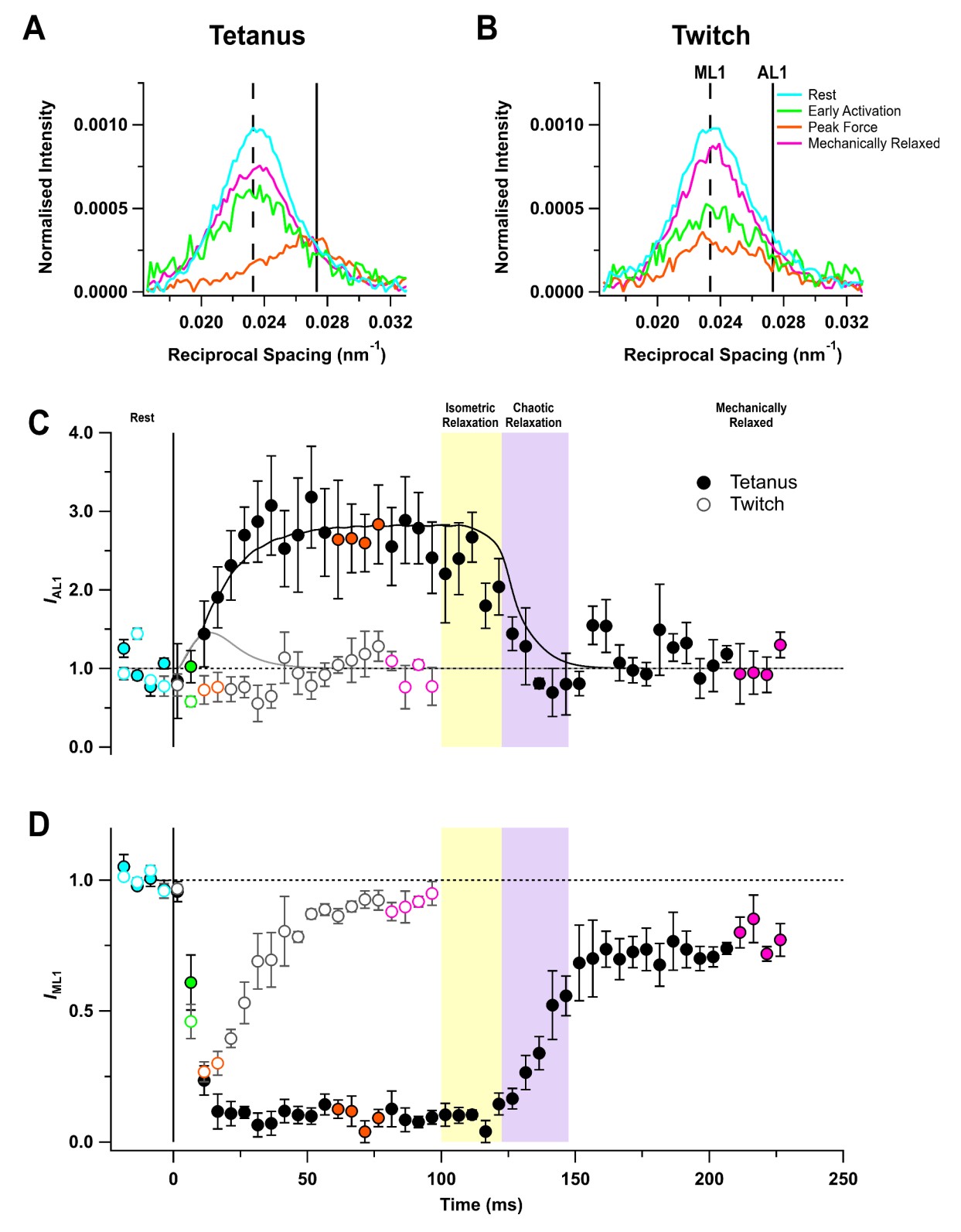

**Figure 3.** Myosin- and actin-based layer line reflections. (**A**) Axial distribution of diffracted intensity in the region of the mixed myosin/actin first layer line in different phases of a tetanus. Cyan, rest; green, early activation; orange, tetanus plateau; magenta, mechanically relaxed. The vertical dashed and continuous lines denote the global best-fit reciprocal spacings of the myosin- and actin-based layer lines, respectively. (**B**) Corresponding results for the twitch, with orange denoting the peak force in the twitch. (**C**) Time course of the intensity of the first actin layer line ($I_{AL1}$) normalised to the

*Figure 3 continued on next page*

*Figure 3 continued*

mean resting value and superimposed on the force in tetanus (black) and twitch (grey). (D) Time course of the intensity of the first myosin layer line ($I_{ML1}$) normalised to the mean resting value. Filled and open symbols in (C) and (D) denote tetanus and twitch, respectively; coloured symbols denote the time periods used to calculate the profiles in (A) and (B); error bars denote SEM for n = 5 muscles for tetanus and n = 4 muscles for twitch. Yellow- and purple-shaded panels denote isometric and chaotic relaxation. Black horizontal dashed lines denote resting values.

The online version of this article includes the following source data for figure 3:

**Source data 1.** An excel file containing the data for individual muscles from which the mean and SEM shown in *Figure 3* were calculated.

signalling not only increased mass at the trigonal positions of the filament lattice occupied by the thin filaments, but also increased disorder.

The ratio of the intensities of (1,1) and (1,0) reflections ($I_{1,1}/I_{1,0}$) signals the movement of myosin motors from the vicinity of the thick filaments towards the thin filaments (*Haselgrove and Huxley, 1973*). $I_{1,1}/I_{1,0}$ was ~0.4 at rest and ~1.8 at the tetanus plateau, similar to values reported previously for mouse EDL (*Caremani et al., 2019*), and ~0.9 at the peak of the twitch (*Table 1*). The increase in $I_{1,1}/I_{1,0}$ was faster than force development at the start of the tetanus (*Figure 2C*, filled circles; *Table 2*), as observed in single fibres from amphibian fast skeletal muscle (*Cecchi et al., 1991*). Since the fraction of myosin motors strongly attached to actin has the same time course as force (*Brunello et al., 2006*), the faster change in $I_{1,1}/I_{1,0}$ shows that myosin motors move towards the thin filaments before they bind strongly to actin or generate force. No clear delay was detected in a single twitch, but this may be due to the low time resolution of the X-ray framing compared to the speed of force development (*Figure 2C*, open circles).

Following tetanic stimulation, $I_{1,1}/I_{1,0}$ decreased during both isometric and chaotic relaxation. However, this change was almost entirely due to a decrease in $I_{1,1}$, and $I_{1,0}$ did not change during isometric relaxation (*Figure 2D*, filled circles), suggesting that myosin motors detached from actin but did not return to the folded 'off' conformation. A small deficit in $I_{1,0}$ with respect to its pre-stimulus value was apparent when mechanical relaxation was complete.

The lateral spacing of the hexagonal lattice of thick and thin filaments ($d_{1,0}$; *Figure 2E*) increased on activation in both the twitch and tetanus, but its time course did not match with that of either force or sarcomere length. Moreover, $d_{1,0}$ did not recover to its pre-stimulus value by the time mechanical relaxation was complete, which may be related to the incomplete recovery of sarcomere order (*Figure 1E*).

## Layer line reflections

The quasi-helical arrangement of myosin motors on the surface of the thick filaments in resting muscle produces a series of off-axial layer line reflections, of which the most intense is the first myosin layer line (ML1), corresponding to a periodicity of 43 nm along the filament axis (*Huxley and Brown, 1967*; *Caremani et al., 2019*; *Ma et al., 2020*). However, the ML1 layer line overlaps the nearby actin-based layer line (AL1) associated with the fundamental helical periodicity of the actin filament, ~37 nm (*ibid*) (*Figure 3A,B*). The AL1 and ML1 reflections were separated by global Gaussian deconvolution of the time series data under the simplifying assumption that their spacings, $S_{AL1}$ and $S_{ML1}$ respectively, do not change during contraction (see 'Materials and methods'). This procedure allowed us to determine the time courses of $I_{AL1}$ (*Figure 3C*) and $I_{ML1}$ (*Figure 3D*). The global best fit values of $S_{AL1}$ and $S_{ML1}$ were 36.6 ± 0.1 (mean ± SE, n = 5) and 43.0 ± 0.1 nm, respectively. $I_{AL1}$ increased by a factor of almost three in the tetanus (*Figure 3C*) with a time course similar to that of force development, but recovered faster than force during isometric and chaotic relaxation, showing that myosin motors detach from actin in both phases.

The profile of the mixed ML1/AL1 layer line showed little change during a twitch (*Figure 3B*), and global deconvolution failed to identify an AL1 component. When $S_{AL1}$ was fixed at the value observed during the tetanus, no increase in $I_{AL1}$ was detected during the twitch (*Figure 3C*, open circles). The intensity of the first myosin layer line ($I_{ML1}$) decreased following the start of stimulation (*Figure 3D*) with a time course much faster than that of force development, showing that the loss of folded helical conformation of myosin motors on activation is much faster than their attachment to actin (*Figure 3D*; *Table 2*). Consistent with this faster response, $I_{ML1}$ decreased almost as much at the peak of the twitch (*Figure 3D*, open circles; *Table 1*) as at the plateau of the tetanus (filled circles).

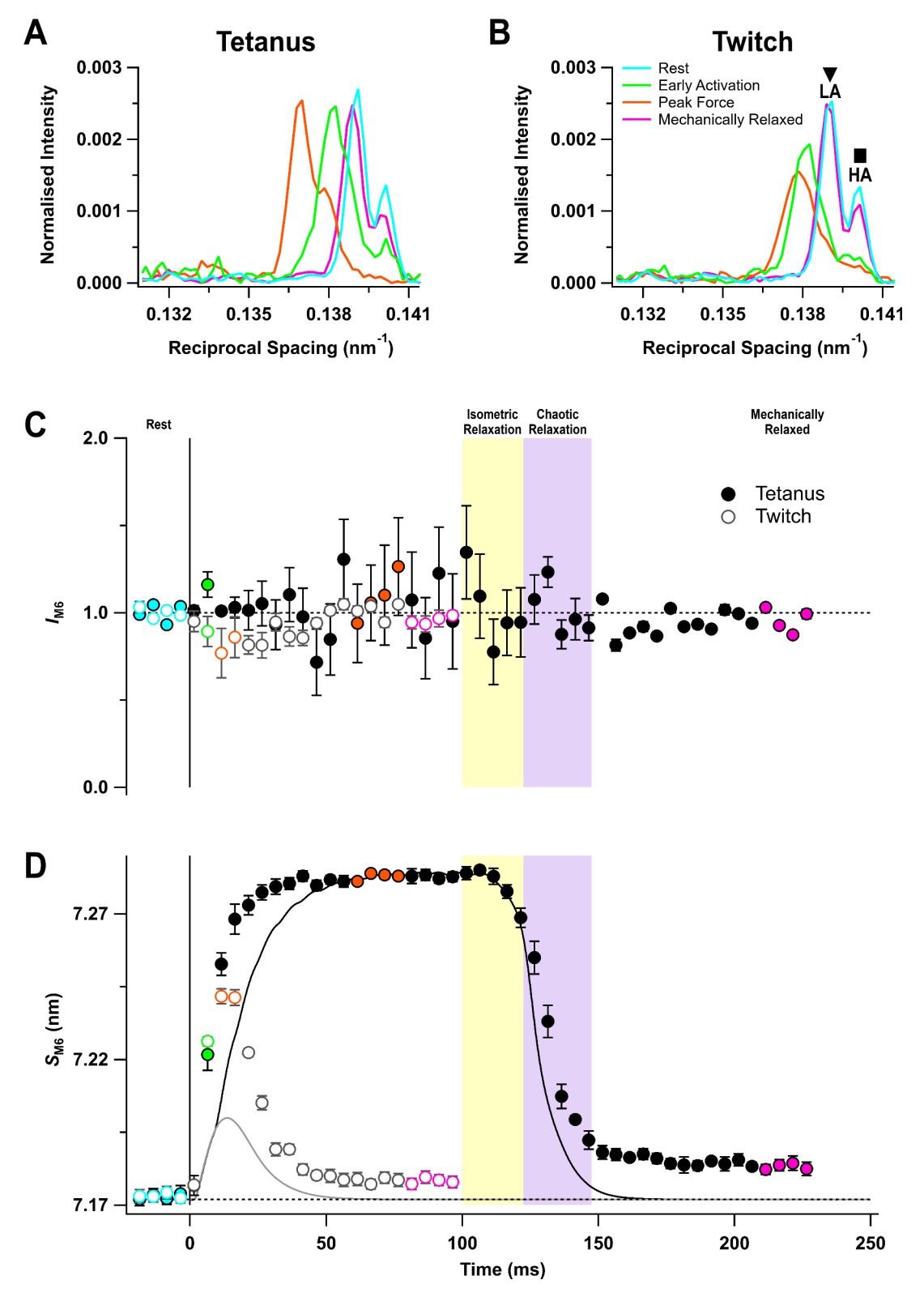

**Figure 4.** The M6 reflection. (A) Axial distribution of diffracted intensity in the region of the M6 reflection in different phases of a tetanus corrected by the cross-meridional width. Cyan, rest; green, early activation; orange, tetanus plateau; magenta, mechanically relaxed. (B) Corresponding results for the twitch, with orange denoting the peak force in the twitch. (C) Time course of the width-corrected intensity of the M6 reflection ($I_{M6}$) normalised to the mean resting value. (D) Time course of the spacing of the M6 reflection ($S_{M6}$) superimposed on the force in tetanus (black) and twitch (grey). Filled

*Figure 4 continued on next page*

*Figure 4 continued*

and open symbols in (C) and (D) denote tetanus and twitch, respectively. Coloured symbols denote the time periods used to calculate the profiles in (A) and (B); error bars denote SEM for n = 5 muscles for tetanus and n = 4 muscles for twitch. Yellow- and purple-shaded panels denote isometric and chaotic relaxation. Black horizontal dashed lines denote resting values.

The online version of this article includes the following source data and figure supplement(s) for figure 4:

**Source data 1.** An excel file containing the data for individual muscles from which the mean and SEM shown in *Figure 4* were calculated.

**Figure supplement 1.** Fractional intensities and spacings of the low-angle and high-angle components of the M6 reflection.

---

$I_{ML1}$ did not recover during isometric relaxation after tetanic stimulation (*Figure 3D*, filled circles), consistent with the above conclusion that myosin motors do not reform the folded helical conformation in this phase, despite the fall of force and detachment of myosin motors from actin. $I_{ML1}$ recovered rapidly but incompletely during chaotic relaxation; the 'off' conformation of the myosin motors characteristic of the resting state had not been recovered even 120 ms after the last stimulus. A smaller recovery deficit was observed after mechanical relaxation in a twitch (open circles).

## Meridional reflections

The meridional region of the diffraction pattern from skeletal muscle is dominated by a series of myosin-based reflections that are generally considered to be of orders of the fundamental ~43 nm axial periodicity of the first myosin layer line, and designated as M1, M2, M3, and so on. Each of these M meridional reflections is in fact a cluster of closely spaced sub-peaks, and in some cases, these can be ascribed to X-ray interference between the two arrays of myosin motors in each thick filament (*Caremani et al., 2021*; *Linari et al., 2000*).

## The M6 reflection

The M6 reflection is associated with a periodic structure in the thick filament backbone with an axial repeat of ~7.2 nm (*Reconditi et al., 2004*; *Huxley et al., 2006*). It has two closely spaced sub-peaks both at rest (*Figure 4A,B*, cyan) and at peak force (orange). The intensity of M6 reflection ($I_{M6}$) was roughly constant during activation, within the reproducibility of the data (*Figure 4C*), but its spacing ($S_{M6}$) increased consistently by 1.54 ± 0.02% at the peak of the tetanus (*Figure 4D*, filled circles; *Table 1*), and by 0.95 ± 0.04% at the peak of the twitch (open circles). This spacing increase is associated with activation of the thick filament (*Haselgrove, 1975*). $S_{M6}$ increased much faster than force at the start of the tetanus (*Table 2*). Following the last stimulus, $S_{M6}$ recovered with the same time course as force during isometric relaxation, but more slowly during chaotic relaxation (*Figure 4D*, filled circles). $S_{M6}$ recovery was still incomplete more than 120 ms after the last stimulus, and there was a smaller recovery deficit following a twitch.

In resting muscle, the lower angle (LA) sub-peak of the M6 reflection is more intense than the higher angle (HA) sub-peak (*Figure 4A,B*), and the relative intensity is similar at the tetanus plateau (*Caremani et al., 2019*; *Figure 4A* – *Figure 4—figure supplement 1B*). The separation between the sub-peaks at rest was 0.0011 nm$^{-1}$, consistent with them being generated by X-ray interference between diffractors in the region corresponding to the first 45 of the 49 layers of myosin motors in each half of the thick filament, counting from its midpoint (*Caremani et al., 2021*). The spacing of the two sub-peaks increased by almost the same amount during the tetanus (*Figure 4—figure supplement 1C*, filled symbols), suggesting that the same diffracting structures are responsible for the M6 reflection during the tetanus. However, at the peak of the twitch (*Figure 4B*, orange) and early during the tetanus rise (*Figure 4A*, green), the HA peak became less intense and the inter-peak separation was larger (*Figure 4—figure supplement 1B,C*).

## The M3 reflection

The M3 reflection is associated with the axial periodicity of the myosin motors along the thick filaments and has been used extensively to obtain information about the conformation and motion of myosin motors during activation, force generation, and relaxation (*Brunello et al., 2009*; *Brunello et al., 2006*; *Reconditi et al., 2011*; *Reconditi et al., 2004*). The intensity of the M3 reflection ($I_{M3}$) decreased transiently in the tetanus before becoming about three times larger than at rest at the tetanus plateau (*Figure 5C*, filled circles; *Table 1*). The initial decrease may be partly due to

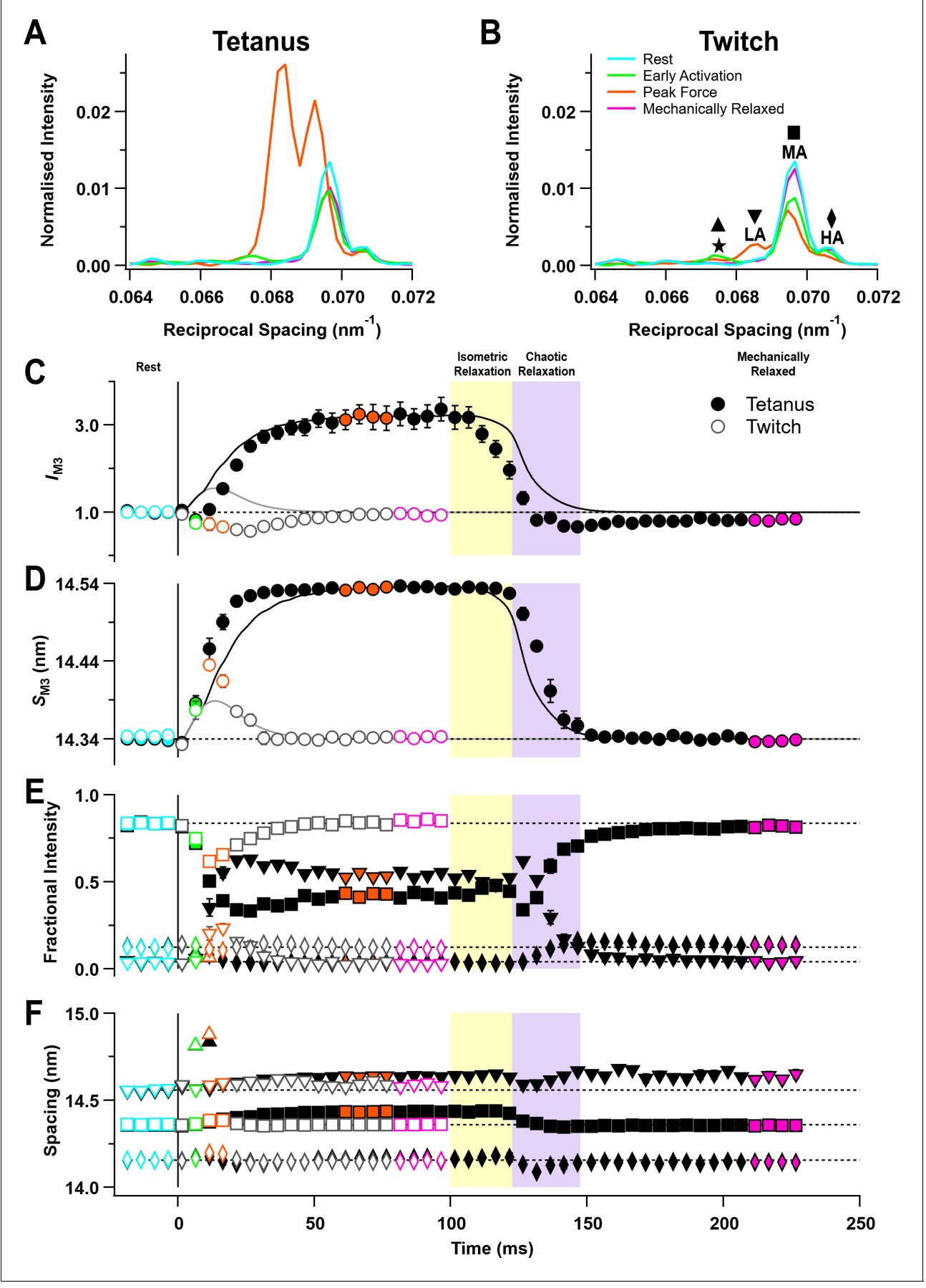

**Figure 5.** The M3 reflection. (**A**) Axial distribution of diffracted intensity in the region of the M3 reflection in different phases of a tetanus corrected by the cross-meridional width. Cyan, rest; green, early activation; orange, tetanus plateau; magenta, mechanically relaxed. (**B**) Corresponding results for the twitch, with orange denoting peak force in the twitch. The star, low-angle (LA), mid-angle (MA), and high-angle (HA) sub-peaks are indicated. (**C**) Time course of the width-corrected intensity of the M3 reflection ($I_{M3}$) normalised to the mean resting value and superimposed on the force in tetanus (black) and twitch (grey). (**D**) Time course of the spacing of the M3 reflection ($S_{M3}$) with force superimposed. (**E**) Fractional intensities of the star (triangles), low-angle (LA; inverted triangles), mid-angle (MA; squares), and high-angle (HA; diamonds) sub-peaks with symbols defined in panel (**B**). (**F**) Spacings of the star, LA, MA, and HA sub-peaks with symbols defined in panel (**B**). Filled and open symbols in (**C–F**) denote tetanus and twitch, respectively. Coloured symbols denote the time periods used to calculate the profiles in (**A**) and (**B**); error bars denote SEM for n = 5 muscles for tetanus and n = 4 muscles for twitch. Yellow- and purple-shaded panels denote isometric and chaotic relaxation. Black horizontal dashed lines denote resting values.

The online version of this article includes the following source data and figure supplement(s) for figure 5:

**Source data 1.** An excel file containing the data for individual muscles from which the mean and SEM shown in *Figure 5* were calculated.

**Figure supplement 1.** Fractional intensities of the sub-peak components of the M3 reflection.

**Figure supplement 2.** Spacings of the sub-peak components of the M3 reflection.

the loss of some of the helically folded myosin motors that were present in the resting muscle and partly due to the destructive interference between the actin-attached and folded populations of motors (*Reconditi et al., 2011*). The slower increase is associated with formation of actin-attached force-generating motors that have their long axes more perpendicular to the filament axis. $I_{M3}$ only decreased in the twitch (open circles).

The spacing of the M3 reflection ($S_{M3}$) increased by 1.36 ± 0.01% at the tetanus plateau and by 0.57 ± 0.05% at the peak of the twitch (*Figure 5D*, *Table 1*), slightly smaller than the corresponding percentage increases in $S_{M6}$ described above. The increase in $S_{M3}$ during the rising phase of the tetanus was faster than force but slower than $S_{M6}$ (*Table 2*). $S_{M3}$ remained remarkably constant during isometric relaxation (*Figure 5D*, filled circles), although $I_{M3}$ recovered substantially towards its resting value (*Figure 5C*), indicating that the motors that detach from actin during isometric relaxation become disordered. $S_{M3}$ decreased rapidly at the onset of chaotic relaxation and had completely recovered to its resting value at the end of mechanical relaxation. $I_{M3}$ had a small but reproducible deficit after the tetanus.

The M3 reflection from resting muscle contains three sub-peaks (LA, MA, HA; *Figure 5B*) with separations that are consistent with X-ray interference between the two myosin head arrays in each thick filament (*Caremani et al., 2021*; *Linari et al., 2000*). The fractional intensity of the LA peak increased during the rise of force in the tetanus (*Figure 5—figure supplement 1C*, filled inverted triangles; *Table 1*), while that of the MA peak decreased (*Figure 5—figure supplement 1D*, filled squares). The time course of these changes was similar to that of $S_{M3}$ but faster than force (*Table 2*). The fractional intensity of the HA peak (*Figure 5—figure supplement 1E*, diamonds) changed much less and was almost the same at the tetanus plateau and at rest. The changes in the fractional intensities of the LA and MA peaks during the twitch (*Figure 5E*, open inverted triangles and squares) were much smaller than those in the tetanus (filled symbols). The spacings of the LA and MA peaks (*Figure 5—figure supplement 2C,D*, inverted triangles and squares) increased during the tetanus, but those of the HA peak (*Figure 5—figure supplement 1E*, diamonds) were the same at the tetanus plateau and at rest. The fractional intensities and spacings of the M3 sub-peaks were almost constant during isometric relaxation at the end of the tetanus (*Figure 5E,F*).

Unexpectedly, a new peak appeared transiently on the LA side of the M3 reflection about 10 ms after the start of stimulation (*Figure 5A,B*, green), with a spacing of 14.832 ± 0.013 nm during the tetanus rise (*Figure 5—figure supplement 2B*, green triangle) and 14.878 ± 0.018 nm at the peak of the twitch (orange triangle). This peak, labelled the 'star' peak in *Figure 5B* (*Caremani et al., 2021*), was not detectable in resting muscle, after mechanical relaxation, or at the tetanus plateau. A peak with a similar spacing was seen previously on cooling of resting EDL muscle of mice and demembranated fibres from rabbit psoas muscle (*Caremani et al., 2021*; *Caremani et al., 2019*), an intervention that mimics activation of the thick filament. The separation between the star and LA peaks was larger than that between the LA and MA or MA and HA peaks (*Figure 5F*).

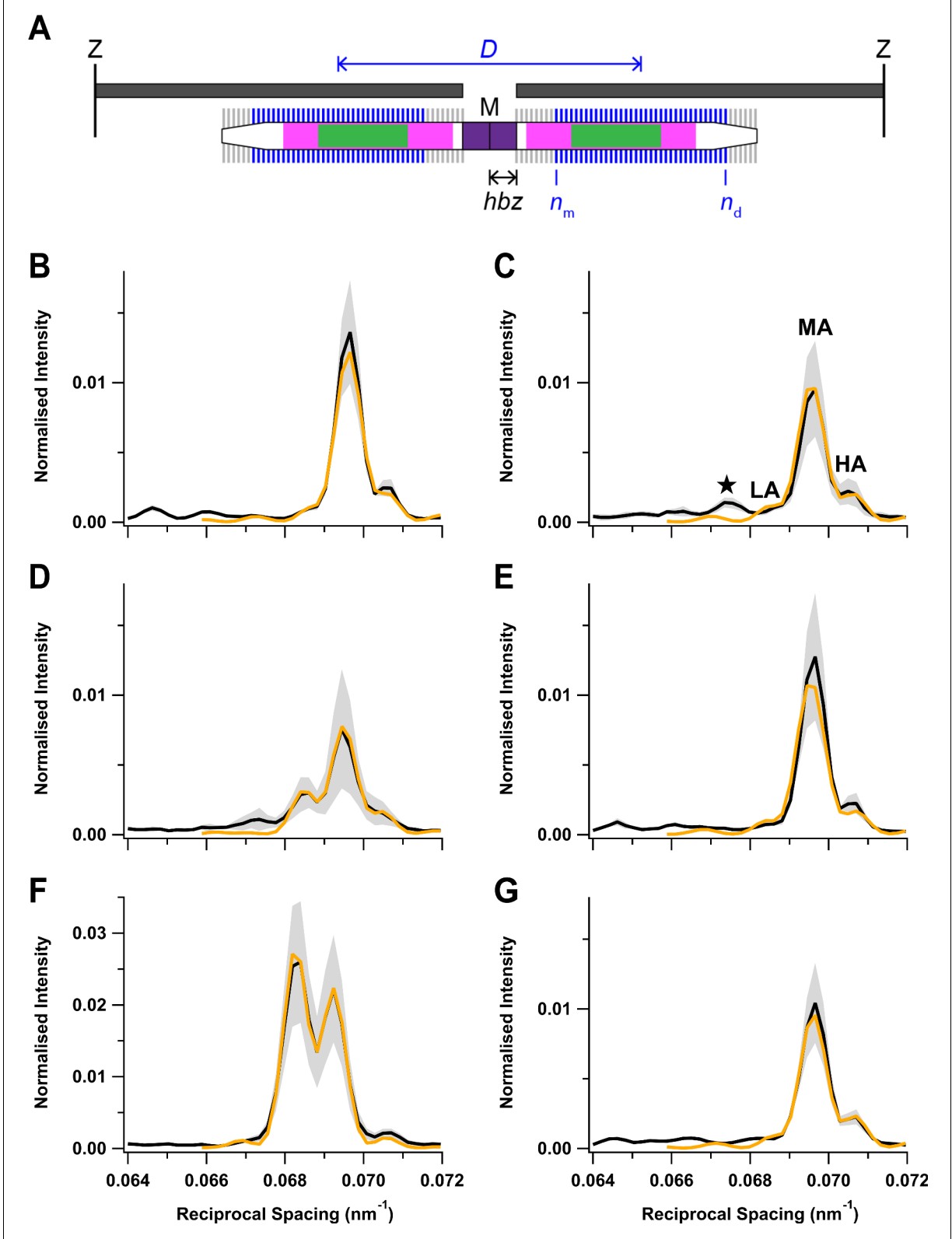

**Figure 6.** Fitting the axial profile of the M3 reflection with a structural model of the thick filament. (**A**) Schematic diagram of the sarcomere for the best-fit model parameters at peak force in the twitch. The sarcomere is delimited by Z-disks (black) and contains overlapping actin (dark grey) and myosin (white) filaments. Myosin filament midpoint, M; layers of myosin motors, vertical lines on myosin filament; bare zone at the centre of the sarcomere lacking myosin motors, purple; half-bare zone, *hbz*; zone of the thick filament containing titin C-type repeats, magenta; myosin-binding protein-C-

*Figure 6 continued on next page*

Figure 6 continued

containing C-zone, green. Ordered layers of myosin motors (blue vertical bars between medial layer $n_m$ and distal layer $n_d$) in the two half-thick filaments have a centre-to-centre or interference distance $D$, shown in blue. Disordered layers are shown in grey. (B–G) Experimental meridional intensity distribution in the region of the M3 reflection (black) with superimposed best fits from the model (orange). Rest (B); early activation (C); twitch peak force (D); twitch mechanical relaxation (E); tetanus plateau (F); tetanus mechanical relaxation (G). Shaded grey areas indicate ± SD from nine (B, C), four (D, E), or five (F, G) muscles.

The online version of this article includes the following source data for figure 6:

**Source data 1.** An excel file containing the data for individual muscles from which the mean and SEM shown in *Figure 6* were calculated.

## Modelling the axial profile of the M3 reflection

The relative intensities and spacings of the sub-peaks of the M3 reflection give information about the axial motion of the diffracting structures—the myosin motors—and their location in the thick filament (*Figure 6A*; *Brunello et al., 2020*). The thick filament is symmetrical about the M-line, and each half-filament contains 49 layers of myosin motors with axial periodicity $d$, with the first layer at a distance $hbz$ (half-bare zone) from the filament mid-point. Only axially ordered myosin motors (blue) from medial layer $n_m$ to distal layer $n_d$ contribute to the diffraction pattern, and the motors in the other layers (grey) are considered to be disordered (see 'Materials and methods' for details). This model gave a good fit (*Figure 6B–G*, orange) to the central region of the M3 profiles containing the LA, MA, and HA sub-peaks, yielding estimates for $hbz$, $d$, $n_m$, and $n_d$ in each condition (*Table 3*). The model did not reproduce the star peak observed during early activation (*Figure 6C*, black) or the other small peaks on the LA side of the M3 reflection at rest (*Figure 6B*), suggesting that they are due to the presence of distinct diffracting structures with a slightly longer axial periodicity (*Caremani et al., 2021*).

As expected from the similarity of the experimental M3 profiles, the best-fit model parameters were similar at rest and following mechanical relaxation after a twitch or tetanus (*Figure 6B,E,G*; *Table 3*). $n_m$ was 2 and $n_d$ close to 46, indicating that nearly all 49 layers of myosin motors were ordered, with the likely exception of three layers at the filament tips. At the tetanus plateau (*Figure 6F*), all layers were ordered; $n_m$ was 1 and $n_d$ was 49. $n_d$ was slightly smaller during early activation and at the peak of the twitch (*Figure 6C,D*), and $n_m$ was larger at the peak of the twitch, indicating that only myosin motors in a central region of each half-filament were ordered (*Figure 6A*), though that region did not correspond to either the conventional 'C-zone' defined by the presence of MyBP-C (green) or the 'C-type' titin repeats (magenta) (*Bennett et al., 2020*; *Caremani et al., 2021*; *Tonino et al., 2019*).

The $hbz$ parameter gives a measure of the centre of mass of the myosin motors with respect to their head-rod junctions and therefore of the motor conformation (*Reconditi et al., 2011*). Motor conformation at the tetanus plateau was previously estimated from experiments on amphibian muscle in which it was perturbed by small lengths or load steps (*Huxley et al., 2006*; *Irving et al., 2000*; *Piazzesi et al., 2007*). Those studies showed that the average centre of mass of the motors at the tetanus plateau was about 3 nm farther from the M line than their head-rod junctions, and it was

**Table 3.** Interpretation of the axial profile of the M3 reflection: best-fit model parameters.

The rows denote the different phases of the tetanus and twitch defined in *Figure 6*. Half-bare zone, $hbz$; medial and distal layers marking the ends of the region of ordered myosin motors, $n_m$ and $n_d$, respectively; axial periodicity between adjacent layers of myosin motors, $d$; intensity scaling factor, $y$; interference distance, $ID$. Mean ± SD from $N$ muscles.

| | N | hbz (nm) | $n_m$ | $n_d$ | d (nm) | y | ID (nm) |
|---|---|---|---|---|---|---|---|
| Rest | 9 | 79.51 ± 0.14 | 2 ± 1 | 46 ± 3 | 14.348 ± 0.010 | 4.5 ± 0.6 | 814 ± 39 |
| Early activation | 9 | 79.40 ± 0.22 | 3 ± 1 | 42 ± 3 | 14.362 ± 0.008 | 4.9 ± 1.0 | 776 ± 53 |
| Twitch PF | 4 | 85.65 ± 0.37 | 9 ± 2 | 43 ± 2 | 14.416 ± 0.013 | 5.1 ± 1.8 | 881 ± 25 |
| Twitch relax | 4 | 79.52 ± 0.12 | 2 ± 1 | 45 ± 3 | 14.348 ± 0.009 | 4.7 ± 0.8 | 790 ± 35 |
| Tetanus PF | 5 | 90.58 ± 0.10 | 1 ± 0 | 49 ± 1 | 14.552 ± 0.003 | 2.0 ± 0.7 | 874 ± 8 |
| Tetanus relax | 5 | 79.61 ± 0.07 | 2 ± 1 | 46 ± 0 | 14.340 ± 0.003 | 5.5 ± 1.3 | 822 ± 12 |

about 8 nm closer to the M line at rest (*Reconditi et al., 2011*). The average movement of the centre of mass of the myosin motors away from the M line on activation was 11 nm, the same as in the present experiments (*Table 3*). However, the best-fit *hbz* values in *Table 3* also show that there was no significant axial motion of the myosin heads during early activation, reflecting the similarity of the intensity of the LA peak at this time point (*Figure 5A,E*, green; *Figure 6C*) to that at rest (*Figure 5A,E*, cyan; *Figure 6B*) and in marked contrast to the large reduction of the helical order of the myosin heads during early activation signalled by $I_{ML1}$ (*Figure 3D*, green). The folded state is completely recovered after mechanical relaxation in a tetanus, again, in contrast to the incomplete recovery of the helical order signalled by $I_{ML1}$. At the peak of the twitch, *hbz* had moved by about 6 nm, about half-way to its tetanus plateau value, suggesting that a significant fraction of motors remain in the folded state in the twitch.

## Discussion

The results presented above allow structural changes in the thick filaments and myosin head or motor domains to be followed with 5 ms time resolution during activation and relaxation of intact mouse EDL muscle. The functional significance of these structural changes is clearer for tetanus than for twitch, in which activation and relaxation are clearly separated by a period of steady-state sarcomere-isometric contraction in which the thin filaments are maximally activated by calcium. Changes in thick filament structure during the tetanus can be correlated with force changes in five sequential phases, which we refer to as activation, tetanus plateau, isometric relaxation, chaotic relaxation, and mechanically relaxed, where the latter is distinct from the resting state achieved several minutes after a previous contraction. Where analogous protocols have been employed, the changes in the thick-filament-based X-ray reflections in these five phases are qualitatively similar to those described previously in fast-twitch amphibian muscles, as described below. The structural mechanisms of thick filament regulation in skeletal muscle have been well conserved across evolution in these species, and this allows the present results to be integrated into the large body of physiological, structural, and mechanical studies on isolated single fibres of amphibian muscle, exploiting the greater homogeneity and lower end-compliance of that preparation. However, most of the published X-ray studies on amphibian muscle fibres were conducted before the significance of thick-filament-based regulation was appreciated; so the synthesis leads to new insights into the underlying mechanisms. No previous muscle X-ray study, to our knowledge, has considered the five phases of the tetanus together or related those phases to the unitary physiological response of skeletal muscle to a single nerve impulse, the twitch.

### Activation

The fastest changes in thick filament structure during activation are the loss of the helical order of the myosin motors, signalled by the ML1 reflection, and the elongation of the filament backbone, signalled by $S_{M6}$. The fraction of molecules in a given conformation is proportional to the amplitude (*A*) of an X-ray reflection, the square root of its intensity. Therefore, the half-time for the loss of the helical order of the myosin motors during the tetanus rise was estimated from the change in $A_{ML1}$, which has a half-time of about 8 ms, the same as that of $S_{M6}$ (*Table 2*). A second group of signals, including the equatorial intensity ratio ($I_{1,1}/I_{1,0}$), the fractional intensity of the LA peak of the M3 reflection ($L_{M3}$), and the axial spacing of the myosin motors ($S_{M3}$), have half-times that are slightly slower, about 11 ms. Moreover, in the early activation time-frame, centred at 6.5 ms after the stimulus, $I_{ML1}$ had reduced by 50% (*Figure 3D*, green), but there was no axial movement of the centre of mass of the myosin motors (*hbz*, *Table 3*), strongly suggesting that the helical order of the myosin motors is lost *before* their release from the folded conformation on the filament backbone. At the time resolution of these experiments, the change in the spacing of the filament backbone ($S_{M6}$) is synchronous with the loss of helical order of the motors, whereas the axial periodicity of the motors ($S_{M3}$) and their radial motion ($I_{1,1}/I_{1,0}$) are synchronous with the loss of the folded population.

The amplitude of the first actin layer line ($A_{AL1}$), signalling myosin motor binding to the thin filaments, has a half-time of 15 ms, similar to that of force. The only X-ray signal that is slower than force is $A_{M3}$ (half-time, 21 ms), and this is likely related to its biphasic response (*Figure 5C*), with a fast decreasing phase followed by a slower rising phase.

This sequence of structural changes during activation is similar to that reported for single fast-twitch fibres from frog skeletal muscle at 4°C (*Brunello et al., 2006*; *Piazzesi et al., 1999*; *Reconditi et al., 2011*), although all the changes are about three times faster in the conditions of the present experiments. Sarcomeres shortened by about 12% during the tetanus rise in these experiments, much larger than the typical 2% in previous studies on single fibres from frog muscle. Sarcomere shortening delays force development and the X-ray signals associated with the loss of the helical order of the myosin motors to the same extent, and the relationship between force and those X-ray signals is unaffected by imposing an additional ~5% shortening at the start of force development (*Linari et al., 2015*). The half-time measured for thick filament activation in the present experiments, about 8 ms, may therefore underestimate the speed of this process under sarcomere-isometric conditions by a few milliseconds, but the above conclusions about the relative time courses of the various structural changes and force development would not be affected by the greater sarcomere shortening in mouse EDL muscle.

The fastest structural changes in the thick filaments, those reported by $A_{ML1}$ and $S_{M6}$, are slower than calcium activation of the thin filaments. The peak of the intracellular $[Ca^{2+}]$ transient is only 2.0 ms after the stimulus in mouse EDL muscle at 28°C, and $Ca^{2+}$ binding to troponin lags this by less than 1 ms (*Baylor and Hollingworth, 2003*). Fluorescent probes on troponin in demembranated fibres from rabbit psoas muscle showed that the rate of calcium-binding in response to a $[Ca^{2+}]$ jump is greater than 1000 s$^{-1}$ at 12°C (*Fusi et al., 2014b*). The half-time of the subsequent azimuthal rotation of tropomyosin around the thin filaments can be estimated by X-ray diffraction from changes in the amplitude of the second actin layer line reflection ($A_{AL2}$), which lies outside the detector in the present experiments. In frog sartorius muscle at 22°C, the half-time of the increase in $A_{AL2}$ was less than 4 ms, about twice as fast as the change in $A_{ML1}$ in those conditions (*Kress et al., 1986*).

Activation of the thick filaments in fast-twitch skeletal muscle, therefore, has two structural components with time courses intermediate between those of thin filament activation and attachment of myosin motors to actin. The delay between thick filament activation and actin attachment is thought to be the result of a structural or biochemical transition in the myosin head in which its catalytic domain becomes strongly bound to actin in a state capable of generating force or filament sliding (*Brenner and Eisenberg, 1987*; *Goldman, 1987*). The delays between thin filament activation and the two components of thick filament activation are less well understood, as is the underlying mechanism of inter-filament signalling (*Irving, 2017*). The transient appearance of the star peak, corresponding to an axial periodicity of 14.85 nm, suggests that this mechanism may be linked to the two closely spaced axial periodicities in the thick filament, the canonical 43 nm periodicity generated by the axial packing of the myosin tails and a longer, ~45.5-nm, periodicity associated with the region of the half-filament associated with a distinct titin-super-repeat in the region of the thick filament containing MyBP-C (*Caremani et al., 2021*). The simple single-periodicity structural model of the thick filament that we used to fit the main part of the M3 reflection (*Figure 6*) only considered the third order of the 43 nm periodicity. The simple model does not reproduce the star peak and does not use the structural information contained in other parts of the X-ray diffraction pattern, including the M6 reflection, the M1 and M2 meridional reflections that would not be produced by a perfect 43 nm helix, and the myosin layer lines. No structural model has yet been developed that can quantitatively reproduce all these features, in any state of a muscle. Such a model would have many more adjustable parameters than the parsimonious model used here, and X-ray data with a higher spatial resolution and signal:noise ratio would probably be required to separate and accurately characterise the interference peaks associated with structural components with different periodicities in distinct zones of the thick filament, in an attempt to constrain those parameters. New synchrotron beam lines and X-ray detectors may allow suitable data to be collected in future studies.

## The tetanus plateau

The thin filaments are fully activated at the tetanus plateau, and about 30% of the myosin motors are likely to be strongly attached to actin (*Caremani et al., 2019*; *Linari et al., 2007*). Modelling the axial profile of the M3 reflection (*Figure 6*; *Table 3*) showed that the average movement of the centre of mass of the myosin motors between the resting state and the tetanus plateau was 11 nm, the same as that reported previously for single fibres from amphibian muscle (*Reconditi et al., 2011*), in which the conformation of the motors at the tetanus plateau had been determined by perturbation

experiments using length or load steps (*Irving et al., 2000*; *Piazzesi et al., 2007*; *Piazzesi et al., 2002*). Given the similarity of sarcomere structure and filament protein isoforms in the two muscle types, it seems very likely that the conformation of both the actin-bound myosin motors at the tetanus plateau and their helical folded conformation in resting muscle is the same in amphibians and mammals.

It is not clear whether there is a residual population of helical or folded motors at the tetanus plateau. The intensity of the ML1 reflection, the X-ray signature of the helical packing of the motors in resting muscle, is about 10% of its resting value (*Figure 3D*; *Table 1*; *Caremani et al., 2019*; *Ma et al., 2018*). Although it is difficult to exclude the possibility that this is due to a population of fibres in the EDL muscle that were not activated by the stimulus, the presence of a similar residual $I_{ML1}$ at the tetanus plateau in single fibres from frog muscle (*Reconditi et al., 2011*) suggests that it may be a general property of fully activated fibres. If the residual $I_{ML1}$ were due to a population of myosin motors remaining in the resting conformation at the tetanus plateau, about one-third of motors would remain in that conformation ($A_{ML1}$ is $0.35 \pm 0.03$; *Table 1*). A slightly higher estimate was obtained from the residual intensity of the fourth myosin layer line in mouse EDL muscle (*Ma et al., 2018*). Since about 30% of motors are actin-attached, and their partners in the same myosin molecule would account for another 30% (*Caremani et al., 2019*; *Piazzesi et al., 2002*), these estimates would account for all the motors, leaving no disordered motors at the tetanus plateau. However, an alternative explanation would be that the region of the myosin motors close to the head-rod junction retains its helical order during activation, although the catalytic domains have lost that order as a result of actin-binding and changes in head conformation. It is also possible that the residual $I_{ML1}$ at the tetanus plateau is associated with another thick filament component that takes up the myosin helical periodicity. Further assessment of those explanations would require further structural modelling constrained by additional in situ structural data.

## Isometric relaxation

Isometric relaxation was first characterised in single fibres from amphibian muscle as a relatively slow, almost linear phase of force decline to about half the tetanus plateau value at a constant sarcomere length after the end of stimulation (*Brunello et al., 2009*; *Edman and Flitney, 1982*). Subsequent studies on isolated myofibrils showed that the rate of isometric relaxation is not limited by the rate of removal of calcium from the intracellular solution but by the isometric rate of detachment of myosin motors from actin (*Poggesi et al., 2005*). In intact muscle fibres, the initial rate of dissociation of calcium from troponin following electrical stimulation is fast (*Caputo et al., 1994*; *Konishi, 1998*) and has been estimated as about 200 s$^{-1}$ in mouse EDL at 28°C (*Baylor and Hollingworth, 2003*). Thin filament inactivation may be slower than this if the remaining actin-bound myosin motors prevent tropomyosin from returning to its off position. In amphibian muscle, the amplitude of the second actin layer line recovers slightly faster than force during early relaxation (*Kress et al., 1986*) but, as discussed by those authors, the difference may be explained by faster inactivation in the segment of the thin filament that does not overlap with thick filaments. Therefore, the available data seem consistent with the hypothesis that the rate of isometric relaxation is limited by that of motor detachment from actin.

Mouse EDL muscles do not show a clear linear phase or 'shoulder' in the time course of mechanical relaxation (*Figure 1C*), and a sarcomere-isometric phase of relaxation has not been reported previously to our knowledge, probably because it is difficult to measure the sarcomere length in these muscles by diffraction of visible light. However, ultra-low-angle X-ray diffraction (*Figure 1*) shows that sarcomeres in mouse EDL muscle *do* remain precisely isometric for the first ~20 ms after the last stimulus (*Figure 1C*), while force decays to about 80% of its plateau value at a progressively increasing rate. $I_{1,1}/I_{1,0}$ and $I_{AL1}$ decrease during isometric relaxation, signalling detachment of myosin motors from thin filaments.

In contrast with the X-ray signals related to detachment of myosin motors from actin, those related to the regulatory state of the thick filament, including $I_{ML1}$, $I_{1,0}$, and $S_{M3}$, and the fractional intensities of the sub-peaks of the M3 reflection do not recover during isometric relaxation; there is no detectable recovery of the helical or folded motor conformations characteristic of resting muscle. The periodicity of the filament backbone ($S_{M6}$) *does* recover partially, tracking the force as expected from its proposed role as a stress sensor (*Linari et al., 2015*). The delay between the loss of actin-

attached myosin motors and the recovery of their ordered resting conformation means that there must be a transient population of disordered motors during isometric relaxation.

## Chaotic relaxation

Sarcomere lengths became inhomogeneous in the period from 20 to 50 ms after the last stimulus of the tetanus (*Figure 1D*). By 50 ms, however, sarcomere uniformity had been regained, sarcomere length had recovered to its resting value, and the force had declined to less than 5% of its tetanic plateau value (*Figure 1C*). Thus, this period from 20 to 50 ms after the last stimulus corresponds to chaotic relaxation as described in amphibian muscle. The X-ray signals associated with actin-attached myosin motors, $I_{1,1}/I_{1,0}$, $I_{AL1}$, and $I_{M3}$, recovered to close to their resting values during chaotic relaxation, although $I_{1,1}/I_{1,0}$ remained slightly higher than its resting value at the end of chaotic relaxation, and $I_{M3}$ slightly lower. The helical order of the motors signalled by $I_{ML1}$ showed a larger recovery deficit, as did the axial periodicity of the filament backbone, $S_{M6}$. The possible origins of these recovery deficits are discussed in the following section.

The overall half-time of the detachment of myosin motors from actin during isometric and chaotic relaxation estimated from $A_{AL1}$, about 23 ms (*Table 2*), or from $I_{1,1}/I_{1,0}$ (30ms), is similar to that of mechanical relaxation itself (27 ms). There was no clear evidence that the rate of detachment estimated from these two X-ray parameters increased on the transition to chaotic relaxation, as might be expected from filament sliding in the latter phase. Recovery of the helical order of the motors during chaotic relaxation, as assessed by $A_{ML1}$, is slower, with a half-time of about 42 ms (*Table 2*), indicating a significant delay between motor detachment and reformation of the helically ordered resting state. This delay is much smaller than that reported for relaxation of mouse soleus muscle at 22°C, in which the half-time of force relaxation was 210 ms, but that for $I_{ML1}$ recovery was 430 ms (*Ma et al., 2020*). Isometric and chaotic relaxation was not resolved after a tetanus in mouse soleus muscle.

## Delayed recovery of the thick filament structure characteristic of resting muscle

The incomplete recovery of $I_{ML1}$ during mechanical relaxation shows that the helical order of the motors characteristic of resting muscle had not recovered when force was close to baseline (*Figure 3*). Only about 80% of motors had regained that conformation as estimated by $A_{ML1}$. Moreover, neither $I_{ML1}$ (*Figure 3D*) nor $S_{M6}$ (*Figure 4D*) recovered further in the remainder of the X-ray sampling period, which ended approximately 130 ms after the last stimulus. In contrast, the axial profile of the M3 reflection and the inferred average centre of mass of the myosin motors had completely recovered at mechanical relaxation (*Figure 6G*; *Table 3*), showing that the myosin motors had folded back against the filament backbone but had not fully reformed the resting helix. Mean sarcomere length had recovered almost to its resting value at the end of chaotic relaxation, but the intensity of the sarcomere reflections remained low (*Figure 1E*), indicating maintained sarcomere heterogeneity. The spacing of the inter-filament lattice ($d_{1,0}$) (*Figure 2E*) had also not recovered to its resting value. The incomplete recovery of the helical packing of myosin motors is unlikely due to sarcomere inhomogeneity or the expanded filament lattice (*Caremani et al., 2021*; *Reconditi et al., 2014*).

The incomplete recovery of helical packing of myosin motors after mechanical relaxation is likely to be the structural correlate of the enhanced twitch force observed after a tetanus, post-tetanic potentiation (*Close et al., 1968*). Myosin motors that have not returned to the helical array would be expected to be recruited more readily by a subsequent stimulus, leading to an enhanced twitch response. Post-tetanic potentiation has been attributed to enhanced phosphorylation of the myosin regulatory light chain (RLC) and, by inference, to release of some myosin motors from the helical array (*Kamm and Stull, 2011*), raising the possibility that the incomplete recovery of the helical array is related to RLC phosphorylation during the tetanus. However, the likely extent of phosphorylation in the present experiments makes this explanation seem implausible. During a 2 s tetanus of mouse EDL muscle at 30°C, RLC phosphorylation increases from about 0.1 to 0.5 mol/mol (*Zhi et al., 2005*). If the extent of phosphorylation is proportional to tetanus duration, as expected for a slow calcium-activated kinase, it would have increased by about 0.02 mol/mol in the present experiments, which seems too small to be responsible for the 20% of myosin motors not returning to the helical array.

An alternative explanation for the incomplete recovery of the helical array after a 100 ms tetanus would be that the structural changes in myosin motors and other thick filament components required to reform this state are intrinsically slow on the 100 ms timescale. However, the observation that the return to the helical array induced by unloaded shortening from the tetanus plateau is already half-complete in about 40 ms in frog muscle fibres at 4°C (*Linari et al., 2015*) argues against this explanation in its simplest form.

Another, perhaps more likely, explanation would be that return of the thick filament to the helical array after a tetanus may be slow because the *thin* filament is not fully off at this time. Several previous studies have reported a slow tail in intracellular free calcium transients following a short tetanus, in both mammalian (*Baylor and Hollingworth, 2003*; *Calderón et al., 2014*; *Hollingworth et al., 1996*) and amphibian fast-twitch muscle fibres (*Caputo et al., 1994*; *Konishi, 1998*). Fast-twitch fibres from both amphibian and mammalian muscles contain millimolar concentrations of the $Ca^{2+}/Mg^{2+}$ exchange buffer parvalbumin, which binds $Mg^{2+}$ in resting muscle. During a tetanus, the $Mg^{2+}$ is replaced by $Ca^{2+}$ (*Hou et al., 1991*), and the slow release of $Ca^{2+}$ from parvalbumin would maintain $[Ca^{2+}]$ above its resting level on the timescale of seconds after the tetanus. Although the magnitude and time course of $[Ca^{2+}]$ and the activation state of the thin filament after short tetani in the conditions of the present experiments are incompletely characterised, it seems plausible that the incomplete recovery of thick filament structure described above might be related at least in part to that of the thin filament.

## Thick-filament control of the strength and dynamics of the twitch

Peak force in a twitch is only about one-quarter of that produced during steady full activation at the tetanus plateau in mouse EDL muscle at 28°C. The X-ray data presented above show that this is primarily due to the lower fraction of myosin motors attached to actin at the peak of the twitch, in comparison with the ~30% of motors attached at the tetanus plateau (*Caremani et al., 2019*). $I_{M3}$ (*Figure 5*) and $I_{AL1}$ (*Figure 3*) actually *decrease* in the twitch, in contrast with their threefold increases at the tetanus plateau. The intensity of the equatorial (1,1) reflection was almost unchanged during the twitch (*Figure 2*), again suggesting a very low fraction of actin-attached motors. The average motor conformation at the peak of the twitch estimated from the axial profile of the M3 reflection (*Table 3*) was also distinct from that at the tetanus plateau, suggesting a small fraction of actin-attached motors combined with a broader range of motor conformations.

The conclusion that the lower force in the twitch is associated with a relatively small number of actin-attached motors is supported by the low number of ATP molecules hydrolysed in a twitch. Heat production in a fixed-end twitch of mouse EDL at 20°C is about 8 mJ/g muscle wet weight, and about half of this is due to the myosin-actin ATPase (*Leijendekker et al., 1987*). This corresponds to about 0.6 ATP molecules hydrolysed per myosin motor, as calculated from the molar enthalpy of phosphocreatine splitting, which buffers ATP hydrolysis in situ, muscle density, and the number of motors per $cm^3$ of the muscle, estimated as $1.04 \times 10^{17}$ for mouse EDL from well-known parameters of muscle ultrastructure (*Barclay et al., 2010*). The mammalian muscle twitch is therefore driven by a fraction of the myosin motors present. The interference fine structure of the M3 reflection suggested that the active motors are preferentially located in the central region of each half filament (*Figure 6A*).

Despite the low force and number of actin-attached motors, the calcium regulatory sites on troponin in the thin filaments are likely to be fully occupied, albeit transiently, at the peak of the twitch. Peak intracellular $[Ca^{2+}]$ is almost the same in the twitch and tetanus, and calcium binding to troponin is fast (*Baylor and Hollingworth, 2003*). In contrast, *thick* filaments are not fully activated at the peak of the twitch. Although the helical array of myosin motors has been disrupted, as signalled by the decreased intensity of the ML1 layer line (*Figure 3*), a significant fraction of motors remain folded back onto the filament surface (*Figure 6*, *Table 3*), making them unavailable for binding to the activated thin filament. The low level of thick filament activation is at least partially responsible for the low fraction of actin-attached motors and force in the twitch.

Although the rate of force development in a twitch, like that in a tetanus, is limited by the rate of thick filament activation and actin attachment, the overall duration of the twitch is primarily determined by the brevity of thin filament activation. The calcium transient has a half-time of only about 2 ms at 28°C. The half-time of $Ca^{2+}$ dissociation from troponin is about 50 ms in mouse EDL at 16°C (*Baylor and Hollingworth, 2003*), likely corresponding to about 25 ms at 28°C, close to the half-

time of force relaxation in the twitch at this temperature (*Table 2*). The intensity of the AL2 layer line associated with tropomyosin movement in a twitch of frog muscle has a half-time of about 36 ms at 24°C, slightly faster than force relaxation (45 ms) in those conditions (*Kress et al., 1986*). Thick filament inactivation in the twitch as monitored by the ML1 layer line has a half-time of about 29 ms (*Table 2*), only slightly slower than force relaxation and *thin* filament inactivation as estimated above. Thus, in contrast with the delayed thick filament inactivation during isometric relaxation after a tetanus, thick and thin filament inactivation in twitch relaxation is almost synchronous.

In summary, the force generated in the twitch—the unitary physiological response of skeletal muscle to action potential stimulation—is partly limited by the incomplete activation of the thick filament, although thin filament activation is transiently complete. The rate of attachment of myosin motors to actin, resolved more clearly in the tetanus, is also partly limited by thick filament activation. Thin and thick filament *inactivation* are rapid and synchronous within the time resolution of current data, and this also contributes to the low fraction of myosin motors attached to actin and peak force in the twitch. This dual-filament description of the regulation of muscle contraction extends the previous narrower focus on structural changes in the thin filament and establishes a physiological structural and functional framework for testing small molecules as potential therapeutics for muscle weakness.

## Materials and methods

### Animals
Male mice (strain C57BL/6J) aged 4–6 weeks were housed at MRC Harwell in groups of four to five in 12:12 hr light:dark cycles at 50% relative humidity, with ad libitum access to water and a standard lab diet. All animals were housed and maintained following the ARRIVE guidelines (*Kilkenny et al., 2010*).

### Muscle preparation
Animals were sacrificed via cervical dislocation in compliance with the UK Home Office Animals (Scientific Procedures) Act 1986, Schedule 1. Following sacrifice, whole EDL muscles were carefully dissected from the hindlimb under a stereomicroscope in a trough continuously perfused with physiological solution (composition in mM: NaCl 118; KCl 4.96; $MgSO_4$ 1.18; $NaHCO_3$ 25; $KH_2PO_4$ 1.17; glucose 11.1; $CaCl_2$ 2.52; pH 7.55, at room temperature) equilibrated with carbogen (95% $O_2$, 5% $CO_2$). Metal hooks were tied with suture silk at the proximal and distal tendons of the muscle to allow attachment to the experimental set-up. The muscle was mounted in a custom 3D-printed plastic chamber between a fixed hook and the lever of a dual-mode force/length transducer (300C-LR; Aurora Scientific, Aurora, Canada), continuously perfused with the physiological solution equilibrated with carbogen at 28°C.

Electrical stimuli were provided by a high-power biphasic stimulator (701C; Aurora Scientific) via parallel platinum electrodes attached to two mylar windows positioned as close as possible to the muscle to minimise the X-ray path in the solution. The stimulus voltage was set to 1.5 times that required to elicit the maximum force response. Muscle length was set to $L_0$, defined as that producing the maximum force in response to a 100 ms train of stimuli at 130 Hz repeated at 5 min intervals. $L_0$ was 12.1 ± 0.4 mm (mean ± SD; n = 7). Muscle cross-sectional area was estimated as (2 x ($W_{MW}$))/($\rho*L_0$), where $\rho$ = 1.06 g.cm$^3$ is the density of the muscle and $W_{MW}$ is the muscle wet weight (*Méndez and Keys, 1960*). $W_{MW}$ was 9.6 ± 0.7 mg, giving a cross-sectional area of 1.50 ± 0.13 mm$^2$.

### X-ray data collection
The trough was sealed to prevent solution leakage and the muscle was mounted vertically at beamline I22 of the Diamond Light Source (Didcot, Oxfordshire, UK) to take advantage of the smaller vertical beam focus to optimise spatial resolution along the meridional axis (*Bordas et al., 1995*; *Huxley et al., 2006*; *Linari et al., 2000*). The monochromatic X-ray beam provided $6 \times 10^{12}$ photons.s$^{-1}$ at 0.1 nm wavelength with full-width at half-maximum about 300 μm horizontally and 100 μm vertically. X-ray diffraction patterns were recorded using a Pilatus P3-2M detector (Dectris Inc, Baden, Switzerland), of an active area of 253.7 × 288.8 mm, with 1475 × 1679 pixels, each 172 × 172 μm, organised in 3 × 8 modules (HxV) with small gaps between modules. The sample-to-

detector distance was set to 8.26 m to optimise the position of the X-ray reflections of interest within the active area of the modules.

For muscle alignment in the X-ray beam, it was attenuated using a 0.1 mm molybdenum attenuator with transmission 0.0056, and regions of each muscle that produced relatively strong muscle-related and weak tendon-related X-ray reflections were identified. Rapid assessment of two-dimensional X-ray patterns was provided by the Data Analysis WorkbeNch software (DAWN; *Basham et al., 2015*).

Following muscle alignment, the attenuator was removed, and time-resolved two-dimensional patterns were collected using either twitch (n = 4 muscles) or tetanic (n = 5 muscles) stimulation at $L_0$. In two of the muscles, both protocols were performed. To minimise radiation damage, X-ray exposure was reduced to a minimum using a shutter and the muscle was moved vertically and/or horizontally between x-ray exposures. Data were acquired with 5 ms time resolution (3 ms acquisition and 2 ms readout time) for 210 ms (42 frames) and 340 ms (68 frames) for twitches and tetani, respectively, with the first 90 ms (18 frames) of each x-ray exposure being required for shutter opening. Four resting frames were acquired before the start of electrical stimulation of the muscle. Signal-to-noise ratio was increased by signal-averaging 8–26 contractions per muscle in the twitch protocol and 6–12 contractions in the tetanus protocol, with the peak force decreasing by 5.0 ± 2.1% in twitches and 12.8 ± 0.5% in tetani between the first and last contraction of the series.

Force, stimulus, muscle length, and X-ray acquisition timing were sampled and generated using the MUSCOLINO software written in LabVIEW (National Instruments) available at https://github.com/LucaFusi-KCL/MUSCOLINO.git (*Fusi, 2021a*, copy archived at swh:1:rev:6b41e429388d98137788624619e9a9ea05a6f93b, *Fusi, 2021b*).

## X-ray data analysis

X-ray diffraction patterns were analysed using DAWN, SAXS package (P. Boesecke; ESRF, Grenoble, France), Fit2D (A. Hammersley; ESRF, Grenoble, France), and Igor Pro 8 (WaveMetrics, Inc). X-ray diffraction patterns containing collagen-based reflections, indicating the presence of tendons in the X-ray beam, were excluded from further analysis. The series of 2D patterns from each contraction was corrected for camera background, added for each muscle, and centred and aligned using the equatorial (1,0) reflections.

The reflections arising from the sarcomere repeats were obtained from unmirrored patterns by integrating from 0.00304 $nm^{-1}$ on either side of the meridian (parallel to the muscle axis). Background intensity distributions were fitted using a convex hull algorithm and subtracted. The intensity and spacing of the sarcomere repeats were determined by fitting multiple Gaussian peaks in the axial region between 0.0033 and 0.0083 $nm^{-1}$. Only even orders of the sarcomere repeat were visible, in agreement with previous findings in amphibian muscle (*Bordas et al., 1987*; *Reconditi et al., 2014*), and were assigned to orders 10–18 in resting muscle (*Figure 1A–B*, cyan). Sarcomere reflections corresponding to orders 8–16 (*Figure 1A*, orange) were observed during a tetanus. Analysis of the sarcomere reflections in *Figure 1D and E* used the two reflections in the axial region between 0.0052 and 0.0072 $nm^{-1}$ in order to avoid regions close to the beam-stop (low-angle side, *Figure 1A–B*), the detector module gap (high-angle side, *Figure 1A–B*), and to a cytoskeletal reflection at 0.0043 $nm^{-1}$ (*Bordas et al., 1987*).

For the analysis of meridional reflections, aligned 2D patterns were mirrored horizontally and vertically. The distribution of diffracted intensity along the meridional axis of the X-ray pattern (parallel to the muscle axis) was calculated by integrating from 0.01522 $nm^{-1}$ on either side of the meridian. Background intensity distributions were fitted using a convex hull algorithm and subtracted; the small residual background was removed using the intensity distribution from a nearby region of the pattern containing no reflections. Integrated intensities were obtained from the following axial regions: M3, 0.066–0.072 $nm^{-1}$; M6, 0.135–0.142 $nm^{-1}$. The cross-meridional width of the M3 and M6 reflections was determined from the radial intensity distribution in the axial regions defined above using a single Gaussian centred on the meridian. The interference components of the M3 and M6 reflections were characterised by fitting multiple Gaussian peaks with the same axial width to the meridional intensity distribution. The total intensity of the meridional reflections was calculated as the sum of the intensity of the component peaks and multiplied by the cross-meridional width to correct for lateral misalignment between filaments during contraction (*Huxley et al., 1982*); the spacing was calculated as the weighted average of that of the component peaks.

For the analysis of the layer line and equatorial reflections, the aligned 2D patterns were mirrored only horizontally and vertically, respectively, to minimise errors in the intensity measurements arising from the presence, on one side of the 2D pattern, of the gaps between detector modules close to reflections of interest. The intensities of the first myosin and first actin layer lines (ML1 and AL1) were calculated by integrating the radial region between 0.0638 and 0.0826 $nm^{-1}$ from the meridional axis. Due to the partial overlap of ML1 and AL1, integrated intensities were obtained from the axial region 0.017–0.033 $nm^{-1}$ for each muscle and separated by global Gaussian deconvolution of the time series data under the simplifying assumption that their spacings, $S_{AL1}$ and $S_{ML1}$ respectively, and axial widths do not change during contraction.

The equatorial intensity distribution was determined by integrating from 0.0036 $nm^{-1}$ on either side of the equatorial axis (perpendicular to the muscle axis), and the intensities and spacings of the (1,0), Z-disk, (1,1), and (2,0) reflections were determined by fitting four Gaussian peaks in the region 0.02–0.065 $nm^{-1}$ with the following constraint: $d_{1,1} = d_{1,0}*(3)^{1/2}$, $d_{2,0} = d_{1,0}*2$, and $d_Z = d_{1,0}*$constant. The latter constraint was necessary for the analysis of time series data because of the overlap between the (1,1) and Z-disk reflections in active muscle.

The four phases of the twitch and tetanus described in the text and figures and used for the M3 modelling are defined with respect to the first stimulus as resting, −18.5 ms to −3.5 ms (four frames, time indicates the centre of the 3 ms data acquisition window); early activation, 6.5 ms (one frame); twitch peak force, 11.5 ms and 16.5 ms (two frames); twitch mechanical relaxation, 81.5 ms to 96.5 ms (four frames); tetanus peak force, 61.5 ms to 76.5 ms (four frames); tetanus mechanical relaxation, 211.5 ms to 226.5 ms (four frames). The time series of 1D intensity distributions for each muscle was normalised to the absolute intensity of the (1,0) reflection at rest to correct for differences in the diffracting mass in the X-ray beam (*Reconditi et al., 2014*). The normalised 1D intensity distributions were then added from four (five) muscles for the twitch (tetanus) protocol, apart from *Figure 6B–C*, which were added from nine muscles.

## Interpretation of the axial profile of the M3 reflection

Myosin filaments from skeletal muscle are centrosymmetric about their midpoint at the M-band of the sarcomere and contain two arrays of 49 myosin motors with an axial periodicity $d$ of about 14.5 nm, corresponding to the spacing of the M3 X-ray reflection ($S_{M3}$), with the first layer (layer 1) starting about 80 nm (the half-bare zone, *hbz*) from the filament midpoint (*Figure 6A*). The centrosymmetric structure of the myosin filament generates interference fringes that sample the X-ray reflection produced by a single array of motors to give finely spaced sub-peaks (*Linari et al., 2000*; Figure 5 of *Brunello et al., 2020*), from which the length of each array of diffracting motors contributing to the M3 reflection and the distance between their centres—the interference distance $D$ (*Figure 6A*)—can be determined. For that purpose, each layer of myosin motors in the myosin filament was represented as a point diffractor. This gives an excellent approximation to the profile of the M3 reflection calculated from the full axial mass profile of the motors (*Reconditi, 2006*).

In the point diffractor representation, *hbz* represents the average centre of mass of the first layer of myosin motors from the filament midpoint, and the diffracted intensity distribution along the meridional axis can be calculated as (*Reconditi, 2006*)

$$I(R) = [\sin(N\pi Rd)/\sin(\pi Rd)]^{2*}[\cos(\pi RD)]^2$$

where $R$ is the reciprocal space co-ordinate, $N$ is the number of layers of motors in a contiguous array from the first medial layer $n_m$ to the distal layer $n_d$ that contributes to the M3 reflection, while layers outside that region make zero contribution because their motors are isotropic, and $D$ is the interference distance calculated as $D = 2*hbz+(n_m +n_d-2)*d$.

For fitting to the experimental M3 profiles, the calculated $I(R)$ was convoluted with a Gaussian function with sigma ~190 μm representing the combined point-spread function of the X-ray beam and detector. The bestfit for each muscle and time point was determined by a global search of *hbz*, $d$, $n_m$, $n_d$, and an intensity scaling factor, $y$, by minimising $\chi^2$ calculated using the experimental standard deviations (*Figure 6B–G*, shaded grey) from the difference between the experimental mean (black) and the model output (orange) for the reciprocal space region 0.066–0.072 $nm^{-1}$, corresponding to 30 detector pixels.

## Statistical analyses

Data in *Figures 1–5* are mean ± SEM. t-tests were used to determine the significance of differences in the half-times of the changes in force and X-ray parameters.

## Acknowledgements

We thank I22 staff, Olga Shebanova, Tim Snow and Nick Terrill (Diamond Light Source), and MRC Harwell for support during the beamtime; Diamond Light Source for the provision of synchrotron beamtime; Marty Rajaratnam (King's College London) for mechanical engineering support and Thomas Kavanagh (King's College London) for helping with 3D printing. This work was funded by the Medical Research Council MR/R01700X/1 and Diamond Light Source. EB and JGO were supported by a British Heart Foundation Intermediate Basic Science Research Fellowship awarded to EB (FS/17/3/32604). LF was funded by a Sir Henry Dale Fellowship awarded by the Wellcome Trust and the Royal Society (210464/Z/18/Z).

## Additional information

### Funding

| Funder | Grant reference number | Author |
| --- | --- | --- |
| Medical Research Council | MR/R01700X/1 | Cameron Hill<br>Malcolm Irving |
| Diamond Light Source | SM21316-1 | Cameron Hill<br>Elisabetta Brunello<br>Luca Fusi<br>Jesús G Ovejero<br>Malcolm Irving |
| British Heart Foundation | FS/17/3/32604 | Elisabetta Brunello<br>Jesús G Ovejero |
| Wellcome Trust | Sir Henry Dale Fellowship 210464/Z/18/Z | Luca Fusi |
| Royal Society | Sir Henry Dale Fellowship 210464/Z/18/Z | Luca Fusi |

The funders had no role in study design, data collection and interpretation, or the decision to submit the work for publication.

### Author contributions

Cameron Hill, Conceptualization, Formal analysis, Investigation, Visualization, Methodology, Writing - original draft, Project administration, Writing - review and editing; Elisabetta Brunello, Luca Fusi, Malcolm Irving, Conceptualization, Formal analysis, Supervision, Funding acquisition, Investigation, Visualization, Methodology, Writing - original draft, Project administration, Writing - review and editing; Jesús G Ovejero, Investigation, Writing - review and editing

### Author ORCIDs

Cameron Hill https://orcid.org/0000-0002-0815-5109
Elisabetta Brunello http://orcid.org/0000-0003-3167-4828
Luca Fusi https://orcid.org/0000-0003-3992-0114
Jesús G Ovejero http://orcid.org/0000-0003-3774-6589
Malcolm Irving https://orcid.org/0000-0001-6419-3583

### Ethics

Animal experimentation: All animals were housed and maintained following the ARRIVE guidelines (Kilkenny et al., 2010; PLOS Biology, 8:e1000412). Animals were sacrificed via cervical dislocation in compliance with the UK Home Office Animals (Scientific Procedures) Act 1986, Schedule 1.

**Decision letter and Author response**
Decision letter https://doi.org/10.7554/eLife.68211.sa1
Author response https://doi.org/10.7554/eLife.68211.sa2

## Additional files

### Supplementary files
• Transparent reporting form

### Data availability
All source data for data analysed in this study are provided for figures 1 to 6.

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
