## [Decision Letter]

**Acceptance summary:**

This study is timely for muscle biophysicists interested in mechanisms of the regulation of muscle myofilament activation and deactivation. This work provides high temporal resolution information on structural changes in myosin, and their role in both thick and thin filament activation.

**Decision letter after peer review:**

Thank you for submitting your article "Myosin-based regulation of twitch and tetanic contractions in mammalian skeletal muscle" for consideration by *eLife*. Your article has been reviewed by 3 peer reviewers, and the evaluation has been overseen by a Reviewing Editor and Anna Akhmanova as the Senior Editor. The following individuals involved in review of your submission have agreed to reveal their identity: Yale E Goldman (Reviewer #2); Kenneth A. Taylor (Reviewer #3).

Essential revisions:

1. Sarcomere shortening in the present conditions is more severe than in the frog muscle preparation, which could delay approach to steady structural configurations. This problem is mentioned, but should be considered more carefully, or explained why it does not influence the observed dynamics.

2. Similarly, Ca^2+^ on and off rates and subsequent kinetics of the thin filament regulatory system should be considered more carefully in regard to kinetics of the x-ray reflections.

3. The model of the thick filament assumes that the structures are homogeneous within each region with sharp borders between regions. But the degree of order and the detailed structural configurations, for instance displacement of the myosin heads from their origins or the proportion of molecules in the folded and helical states within a zone, and other structural parameters, might not be perfectly constant within each region. This kind of complexity would be difficult to constrain from the data, so the parsimonious model, with minimal parameters, is necessary. However, the potential influence on conclusions of possible more complex structural distributions should be considered.

4. Figure 4 A,B and C are visually inconsistent; Figure 4A shows the M6 intensity at the selected time points of a tetanus as different but Figure 4C shows the M6 intensity as constant at literally all points. The figure needs to be revised to show why the intensity does not change. A textual explanation is not sufficient because the difference is so large.

*Reviewer #1:*

In this manuscript the authors provide high temporal and spatial resolution of the sequence of events for myosin structural changes during twitch and tetanus contractions and during relaxation. They provide evidence that thick filaments are not fully activated during a twitch contraction and that there are differences in how fast myosins can resume there resting positions on the thick filament backbone for twitch versus tetanic contraction of a fast twitch mammalian muscle The data also allow estimations of populations shifts of myosin states, e.g. resting disordered and actin bound, in different contraction and relaxed conditions. The data are compelling and support their conclusions and provide new insights into the coordinated activity of think and thin filaments during contractile activation, steady tension and relaxation.

1. Lines 32 and 42. The statement that Ca^2+^ regulatory sites are fully occupied and thin filaments are completely activated are at odds with the conclusions drawn in the discussion that thin filaments are not completely activated or deactivated due to the lower numbers of bound myosin during a twitch vs. tetanus contraction.

2. Lines 56-58. Repeated sentence – one should be removed

3. Lines 114-118. The point being made here is not clear.

4. Some conclusions regarding the kinetics of thin filament activation and deactivation would be strengthened by manipulations that alter myosin recruitment and/or thin filament calcium binding.

*Reviewer #2:*

The authors performed detailed x-ray diffraction measurements during twitches and tetani on whole mouse EDL muscles at the Diamond Light Source synchrotron. The instrumentation enabled them to signal average diffraction peak intensities from successive contractions giving 5 ms effective time resolution for structural changes primarily in the myosin motor domains and thick filament backbone. The detector also captured lower angle peaks interpreted as sarcomere length changes. The experiments were done with care and described clearly.

Much of the results largely confirm earlier experiments of similar type in single frog muscle fibers, but having the extensive simultaneously measured dynamics on a mammalian muscle reported here is useful.

Factors that might influence the interpretation could be discussed more thoroughly. Sarcomere shortening in the present conditions is more severe than in the frog muscle preparation which could delay approach to steady structural configurations. Similarly calcium and thin filament structural dynamics are mentioned only briefly.

The model of the thick filament used assumes that the structures are homogeneous within each region with sharp borders between regions. But the degree of order and the detailed structural configurations, for instance displacement of the myosin heads from their origins or the proportion of molecules in the folded and helical states within a zone, and other structural parameters, might not be perfectly constant along each region. This kind of complexity would be difficult to constrain from the data, so the parsimonious model, with minimal parameters, is necessary. However, the potential influence on conclusions of possible more complex structural distributions should be considered.

Several observations are left for further studies, including a previously observed, but unexplained meridional reflection near the sampled M3 peaks and delayed return of several reflections to their pre-stimulus values.

This paper provides an extensive, reliable dataset for considering control and dynamics of mammalian muscle contraction.

Several of the reflections do not return to their pre-stimulus values promptly after the contraction. They return eventually as is evidenced by the difference between pre-stimulus baseline and the after-deficit. The time course of these return kinetics and how they depend on intervening conditions, such as temperature and extra stimuli are not treated at all. The main stimuli were repeated at 5 min intervals, but how long the recovery takes within this time is left open. There is a lot of speculation about phosphorylation of RLC or residual Ca^2+^ in the cytoplasm, no substantive data are presented on these ideas.

*Reviewer #3:*

Sarcomere length disordering during relaxation was previously described from observations in vertebrate striated muscle but mostly from amphibians. This report provides equivalent measurements from mammalian muscle and improves on previous measurements in which the sarcomere length was measured by laser diffraction simultaneous with those structures probed by the X-ray diagram. This report measured sarcomere length from the set of very low angle X-ray reflections that come from the sarcomere length itself. Thus, the high angle measurements describing the changes in the thick filament and the myosin heads and the low angle measurements providing the sarcomere length all come from the same region of the muscle at the same time using the same irradiation. The authors have also carefully controlled temperature which has a disordering effect on the myosin head arrangement in mammalian muscle. Consequently, more insightful interpretations are possible from these data.

Decades of study of the X-ray diffraction from mammalian striated muscle has established what information is reported by many of the key reflections. The equatorial 1,0 reflection generally reports mass at the thick filament position, and the 1,1 generally reports mass at the thin filament position; when the 1,0 reflection goes down, the 1,1 reflection goes up showing mass movement toward the thin filament. If tension rises when this occurs, it means myosin heads are binding actin strongly and producing tension. The myosin first layer line reports the ordering of myosin heads around the thick filament, the actin first layer line reports myosin heads binding actin. The M6 meridional reflection reports on homogeneity of the 14.3 nm periods on the thick filament and changes in this average spacing, which generally are believed to indicate activation of the thick filament. This paper puts the relative timing of these changes on a firmer footing as well as establishes that relaxation following a tetanus occurs slowly, thereby leaving the myosin heads in a potentiated state where a subsequent contraction can be produced more quickly.

The changes in the M6 reflection are somewhat unexpected. The plots showing the axial profile show the normalized peak intensity dropping but the width does not change much, yet the integrated intensity appears to remain rather constant throughout the whole process. If the muscle shortens, expectation is that the increase in the number of repeats brought about by the shortening should increase the integrated intensity if there is a large enough shortening. If the peak intensity drops, there should be an increase somewhere else to keep the integrated intensity constant, radial broadening (?), to compensate yet the graphs do not seem to bear this out. The spacing increase occurs quickly and precedes force development, further attesting to the early observations that the spacing change was due to activation, not tension development. The novel observation of this experiment is that recovery to the resting value is very slow, not occurring even 120 msec following recovery to resting tension. The time constant of the relaxation phase is very different from the activation phase, thus leaving the muscle in a potentiated state.

There are plenty of novel observations in this manuscript which should serve as good food for thought toward an understanding of striated muscle contraction.

Data availability: The authors provide Excel spread sheets giving the intensity data etc. The actual X-ray videos themselves are not provided. However, I believe there is no public depository for such data that I am aware of, though there should be. Possibly, the BioCAT (Tom Irving) is making progress on creating such a depository.

When the reader sees Figure 4A,B, the change in peak height and spacing is their first impression while Figure 4C and the text suggests no change in intensity. It is quite unbelievable that the red curves and the cyan curves in Figure 4A,B have the same integrated intensity; they have very different peak intensities. It might be a good idea to clarify that the peak intensity falls, but the integrated intensity remains constant if indeed that is what is meant. Since the integrated intensity would be a function of the number of diffracting entities, the peak intensity would be a measure of how similar in spacing these entities are. The myosin heads assume very different structures during relaxation and activation, some so disordered that they don't contribute to the intensity of reflections that report myosin head structure. The M6, however, is reporting on the backbone structure, which due to the packing of myosin coiled-coil domains, has far less freedom for structural change other than to change length either homogeneously or heterogeneously. Heterogeneity in the spacing would reduce the peak intensity and increase the axial width of the reflection. Being a skeptical reviewer, I cannot fathom that the graphs shown in Figure 4A,B have the same intensity as attested to in Figure 4C. They have very different peak heights and roughly similar widths, yet they are said to have the same integrated intensity. I think the authors would be well advised to be careful in these comparisons, perhaps specifying peak intensity and integrated intensity in those instances where the difference might be important. I don't believe this is a niggling point. One of the main points of the paper is the thick filament tension sensing, which is a backbone issue. It may be that a more sophisticated graphical display, perhaps a 3-D graph plotting axial, radial coordinates in x and y and intensity in z for the four states would be more convincing.

The changes in the M3 reflection are also somewhat confusing. Presumably, the M3 and M6 are reporting on the spacings in the filament itself due to the arrangement of myosin molecules, yet, the M3 spacing change (1.36%) is less than the M6 spacing change (1.54%). Why is this? The M3 reports on the heads in the relaxed muscle are already irregularly spaced, but the average is the same fundamental period. How can the heads get out of sync with the tails to which they are attached? I am not sure this can be blamed on the non-myosin proteins.

---

## [Author Response]

Essential revisions:1. Sarcomere shortening in the present conditions is more severe than in the frog muscle preparation, which could delay approach to steady structural configurations. This problem is mentioned, but should be considered more carefully, or explained why it does not influence the observed dynamics.

Shortening during force development delays both force development itself and the associated structural changes in the thick filaments. The magnitude of the delay can be estimated from the effects of imposing an additional ca 5% shortening at the start of force development in isolated frog muscle fibres (Brunello et al., 2006; Linari et al., 2015), in which the central sarcomeres shorten by typically only 2% during fixed end contractions, compared with the ca 12% in the mouse EDL muscles. Those authors found that imposition of an additional ca 5% shortening at the start of force development produced the same delay, within the precision of the measurements, in force and the activation state of the thick filament as determined by the spacing of the M6 reflection (*S*_M6_). In fact, there was a unique relationship between force and *S*_M6_ in the two protocols (Linari et al., 2015). Therefore, although sarcomere shortening does delay the structural changes reported here, it does so by the same extent as force, so conclusions related to the relative time courses are unaffected. The section of the Discussion at lines 436 to 442 of the resubmitted manuscript has been expanded to clarify and quantify this point.

2. Similarly, Ca^2+^ on and off rates and subsequent kinetics of the thin filament regulatory system should be considered more carefully in regard to kinetics of the x-ray reflections.

The kinetics of the structural changes reported by the X-ray reflections were compared quantitatively with those of the intracellular [Ca^2+^] transient in the same preparation and at the same temperature in lines 444-454 of the resubmitted manuscript. The same paragraph described the quantitative relationship with the ON rates for troponin binding as calculated from in vitro rate constants, and the in situ rate determined using caged calcium experiments with probes on troponin. The rate of the subsequent azimuthal motion of tropomyosin determined in situ from the second actin X-ray layer line was also discussed and quantitatively compared with the kinetics of the myosin-based layer lines in the same paragraph. Similar quantitative comparisons with relevant published in vitro and in situ measurements of the corresponding OFF rates (including experiments with isolated myofibrils in which it is possible to make fast decreases in [Ca^2+^]) were made in the three paragraphs, now on lines 525-542 (isometric relaxation phase), lines 628-641 (incomplete recovery of the OFF structure after mechanical relaxation) and lines 682-695 (summary of the relative kinetics of thin and thick filament activation and inactivation) of the resubmitted manuscript. We are unaware of additional published data that could significantly further illuminate those extensive quantitative comparisons in the conditions of our experiments, although substantive empirical uncertainty remains (well recognised in the literature and discussed on lines 628-641 of the resubmitted manuscript) about a possible slow tail of the intracellular [Ca^2+^] transient.

3. The model of the thick filament assumes that the structures are homogeneous within each region with sharp borders between regions. But the degree of order and the detailed structural configurations, for instance displacement of the myosin heads from their origins or the proportion of molecules in the folded and helical states within a zone, and other structural parameters, might not be perfectly constant within each region. This kind of complexity would be difficult to constrain from the data, so the parsimonious model, with minimal parameters, is necessary. However, the potential influence on conclusions of possible more complex structural distributions should be considered.

As the reviewer says, we chose a parsimonious model with parameters that could be constrained by the observed profiles of the M3 reflection at key phases of the experimental protocols used here (Figure 6). Some limitations of the simple model originally discussed on ll. 444-449 of the submitted manuscript are now expanded upon on lines 460-474 of the resubmitted manuscript, and more extensively by Caremani et al. (2021), who used the same model. There are two fundamental limitations of the model, in our view, which are related to the lack of a high-resolution structure of a vertebrate thick filament in any state. One is that the simple model does not account for the presence of multiple periodicities in the thick filament (Caremani et al., 2021), with its empirical correlate, the presence of additional sub-peaks that don’t index on the fundamental *d_m_* periodicity. The other is that the model is limited to fitting the M3 reflection, and does not use the information in the other reflections, in particular the M6 reflection from the thick filament backbone, the ‘forbidden’ reflections M1 and M2, and the myosin-based layer lines. At present there is no structural model that can reproduce all these features, in any state of a muscle. However the increased spatial resolution and sensitivity available at new synchrotron beam lines with the latest detectors seems to offer the possibility to separate the interference peaks associated with the structural components in distinct zones of the thick filament with two closely-spaced periodicities, which may allow such more complete models to be constrained in future studies. These considerations have now been added to the Discussion of the present paper.

4. Figure 4 A,B and C are visually inconsistent; Figure 4A shows the M6 intensity at the selected time points of a tetanus as different but Figure 4C shows the M6 intensity as constant at literally all points. The figure needs to be revised to show why the intensity does not change. A textual explanation is not sufficient because the difference is so large.

The discrepancy between panels A,B and C of Figure 4 was due to *I*_M6_ in Figure 4C being corrected by the cross-meridional width of the reflection, whereas the axial profiles in A and B were not corrected for differences in cross-meridional width. We have now removed the confusion by replacing the profiles in Figure 4 A and B by width-corrected profiles. For consistency, we have done the same for the M3 reflection profiles in Figure 5 A and B.

Reviewer #1:[…] 1. Lines 32 and 42. The statement that Ca^2+^ regulatory sites are fully occupied and thin filaments are completely activated are at odds with the conclusions drawn in the discussion that thin filaments are not completely activated or deactivated due to the lower numbers of bound myosin during a twitch vs. tetanus contraction.

The relevant part of the Discussion (ll. 624-627 of the submitted version) stated:

“Despite the low force and number of actin-attached motors, the thin filaments are fully activated, albeit transiently, at the peak of the twitch. […] In contrast, *thick* filaments are not fully activated at the peak of the twitch.” This statement is fully consistent with those on ll. 32 and 42 of the Introduction. However, it implicitly considers that the activation state of the thin filament is uniquely defined by the calcium occupancy of troponin. The reviewer is probably thinking of thin filament regulation models in which their activation state is additionally determined by the extent of myosin binding. Because the X-ray signals in the present paper only give structural information about myosin motors and filaments, they do not give new information about the possible complexity of thin filament activation. Therefore, we considered only the simplest model for thin filament activation in this paper, but we have now edited this section of the Discussion on lines 671-672 to remove our explicit assumption of that simplest model.

2. Lines 56-58. Repeated sentence – one should be removed.

This sentence has now been deleted.

3. Lines 114-118. The point being made here is not clear.

We expanded this section (lines 115-123 of the resubmitted manuscript) to clarify that nearly all the published studies of thin filament regulation, for example those using pCa titrations with steady state Ca buffering in skinned muscle fibres, have been made at relatively low temperature in order to minimise the irreversible effects of sustained high levels of activation at high temperature. However, the role of thick filament regulation was inadvertently excluded from such experiments because the thick filaments are already switched on at low temperature, even at pCa 9.

4. Some conclusions regarding the kinetics of thin filament activation and deactivation would be strengthened by manipulations that alter myosin recruitment and/or thin filament calcium binding.

The focus of the present paper is on *thick* filament-based regulation, but we agree that fundamental questions remain about *thin* filament regulation and the coupling between the regulatory states of the thin and thick filaments, in particular for the inactivation or relaxation phase. Mechanical and pharmacological perturbations or the use of muscles from transgenic mouse models may be informative, and these may be the subjects of future studies.

Reviewer #2:The authors performed detailed x-ray diffraction measurements during twitches and tetani on whole mouse EDL muscles at the Diamond Light Source synchrotron. The instrumentation enabled them to signal average diffraction peak intensities from successive contractions giving 5 ms effective time resolution for structural changes primarily in the myosin motor domains and thick filament backbone. The detector also captured lower angle peaks interpreted as sarcomere length changes. The experiments were done with care and described clearly.Much of the results largely confirm earlier experiments of similar type in single frog muscle fibers, but having the extensive simultaneously measured dynamics on a mammalian muscle reported here is useful.Factors that might influence the interpretation could be discussed more thoroughly. Sarcomere shortening in the present conditions is more severe than in the frog muscle preparation which could delay approach to steady structural configurations. Similarly calcium and thin filament structural dynamics are mentioned only briefly.The model of the thick filament used assumes that the structures are homogeneous within each region with sharp borders between regions. But the degree of order and the detailed structural configurations, for instance displacement of the myosin heads from their origins or the proportion of molecules in the folded and helical states within a zone, and other structural parameters, might not be perfectly constant along each region. This kind of complexity would be difficult to constrain from the data, so the parsimonious model, with minimal parameters, is necessary. However, the potential influence on conclusions of possible more complex structural distributions should be considered.Several observations are left for further studies, including a previously observed, but unexplained meridional reflection near the sampled M3 peaks and delayed return of several reflections to their pre-stimulus values.This paper provides an extensive, reliable dataset for considering control and dynamics of mammalian muscle contraction.

Most of these points have been addressed under the first three points of ‘Essential revisions’ above. There is an additional point here about the novelty of the results in relation to previous studies on amphibian muscle. In that regard we would like to point out that although, where direct comparisons can be made, results from mammalian and amphibian muscles are qualitatively similar, there is to our knowledge no published X-ray study in amphibian muscle in which twitch and tetanic contractions are compared directly. The relatively low twitch:tetanus force ratio in mammalian muscles at physiological temperature makes this distinction significant.

Several of the reflections do not return to their pre-stimulus values promptly after the contraction. They return eventually as is evidenced by the difference between pre-stimulus baseline and the after-deficit. The time course of these return kinetics and how they depend on intervening conditions, such as temperature and extra stimuli are not treated at all. The main stimuli were repeated at 5 min intervals, but how long the recovery takes within this time is left open. There is a lot of speculation about phosphorylation of RLC or residual Ca^2+^ in the cytoplasm, no substantive data are presented on these ideas.

We intend to investigate the time course of recovery of the fully OFF structure of the thick filament after a tetanus in future studies, and it may be informative to compare that time course with that of dephosphorylation of the RLC. However, the rate of RLC phosphorylation *during*tetanic stimulation in mouse EDL muscle would lead to only 2% of the RLCs being phosphorylated in the protocol used in our experiments (Zhi et al., 2005). Therefore, as argued on lines 564-578 of the submitted manuscript (lines 605-619 in the resubmitted manuscript), alternative explanations for the incomplete recovery of thick filament structure after a short tetanus should also be explored.

Reviewer #3:[…] Data availability: The authors provide Excel spread sheets giving the intensity data etc. The actual X-ray videos themselves are not provided. However, I believe there is no public depository for such data that I am aware of, though there should be. Possibly, the BioCAT (Tom Irving) is making progress on creating such a depository.

We will be happy to provide the time-resolved X-ray data upon request, and to deposit them in a suitable repository should one become available.

When the reader sees Figure 4A,B, the change in peak height and spacing is their first impression while Figure 4C and the text suggests no change in intensity. It is quite unbelievable that the red curves and the cyan curves in Figure 4A,B have the same integrated intensity; they have very different peak intensities. It might be a good idea to clarify that the peak intensity falls, but the integrated intensity remains constant if indeed that is what is meant. Since the integrated intensity would be a function of the number of diffracting entities, the peak intensity would be a measure of how similar in spacing these entities are. The myosin heads assume very different structures during relaxation and activation, some so disordered that they don't contribute to the intensity of reflections that report myosin head structure. The M6, however, is reporting on the backbone structure, which due to the packing of myosin coiled-coil domains, has far less freedom for structural change other than to change length either homogeneously or heterogeneously. Heterogeneity in the spacing would reduce the peak intensity and increase the axial width of the reflection. Being a skeptical reviewer, I cannot fathom that the graphs shown in Figure 4A,B have the same intensity as attested to in Figure 4C. They have very different peak heights and roughly similar widths, yet they are said to have the same integrated intensity. I think the authors would be well advised to be careful in these comparisons, perhaps specifying peak intensity and integrated intensity in those instances where the difference might be important. I don't believe this is a niggling point. One of the main points of the paper is the thick filament tension sensing, which is a backbone issue. It may be that a more sophisticated graphical display, perhaps a 3-D graph plotting axial, radial coordinates in x and y and intensity in z for the four states would be more convincing.

The discrepancy between panels A,B and C of Figure 4 was due to *I*_M6_ in Figure 4C being corrected by the cross-meridional width of the reflection, whereas the axial profiles in A and B were not corrected for differences in cross-meridional width. We have now removed the confusion by replacing the profiles in A and B by width-corrected profiles. For consistency, we have done the same for the M3 reflection profiles in Figure 5A and B.

The changes in the M3 reflection are also somewhat confusing. Presumably, the M3 and M6 are reporting on the spacings in the filament itself due to the arrangement of myosin molecules, yet, the M3 spacing change (1.36%) is less than the M6 spacing change (1.54%). Why is this? The M3 reports on the heads in the relaxed muscle are already irregularly spaced, but the average is the same fundamental period. How can the heads get out of sync with the tails to which they are attached? I am not sure this can be blamed on the non-myosin proteins.

In general the observed M3 and M6 spacings are not related by exactly a factor of two, e.g. in their time course during activation and relaxation, in both skeletal (Brunello et al. 2006) and heart muscle (Brunello et al. 2020), and in their sarcomere length dependence in skeletal muscle (Reconditi et al. 2014), and temperature-dependence in skeletal muscle (Caremani et al., 2019, 2021). One possible explanation is that different regions of the thick filament (e.g. the C zone vs the D zone) have slightly different axial periodicities and their relative contribution to a given reflection is not constant (Brunello et al., 2020).